# Optimizing structured surfaces for diffractive waveguides

Yuntian Wang [1,2,3,4], Yuhang Li [1,2,3,4], Tianyi Gan[1,3], Kun Liao [1,2,3], Mona Jarrahi [1,3] & Aydogan Ozcan [1,2,3] ✉

We introduce universal diffractive waveguide designs that can match the performance of conventional dielectric waveguides and achieve various functionalities. Optimized using deep learning, diffractive waveguides can be cascaded to form any desired length and are comprised of transmissive diffractive surfaces that permit the propagation of desired modes with low loss and high mode purity. In addition to guiding the targeted modes through cascaded diffractive units, we also developed various waveguide components and introduced bent diffractive waveguides, rotating the direction of mode propagation, as well as spatial and spectral mode filtering and mode splitting diffractive waveguide designs, and mode-specific polarization control. This framework was experimentally validated in the terahertz spectrum to selectively pass certain spatial modes while rejecting others. Without the need for material dispersion engineering diffractive waveguides can be scaled to operate at different wavelengths, including visible and infrared spectrum, covering potential applications in, e.g., telecommunications, imaging, sensing and spectroscopy.

Optical waveguides are indispensable elements in optics and photonics, essential for the precise control and transmission of light for a variety of applications, including, e.g., telecommunications[1–4], sensing[5–8], and integrated photonic circuits[9–11]. These waveguides confine light through a medium with a higher effective refractive index surrounded by regions with a lower effective refractive index, enabling efficient light propagation with minimal loss. There are several types of waveguides tailored to specific applications. For example, planar waveguides, consisting of a thin film of a high-refractive-index material deposited on a substrate with a lower refractive index, confine light within the film and allow propagation parallel to the substrate plane[12–14]. This design is particularly useful in integrated photonic circuits due to its compatibility with lithography processes[11,15]. Strip waveguides, also known as rib or channel waveguides, involve etching a strip or ridge of high-refractive-index material onto a substrate, enhancing light confinement and making it suitable for compact photonic circuits[15–17]. Fiber optic waveguides comprise a core of a

higher refractive index surrounded by a cladding of a lower refractive index, ensuring total internal reflection for efficient long-distance light transmission[18,19]. Additionally, other waveguide components like mode filters and mode splitters are crucial for mode selection[20–22] and spatial demultiplexing[23–25], respectively. To achieve these goals, traditional waveguide designs require careful consideration of various factors such as refractive index contrast, material dispersion, waveguide dimensions, and structures[26,27]. Furthermore, advanced fabrication techniques such as chemical vapor deposition[28–30], lithography[31,32], and thermal drawing[33,34] are typically required for these waveguide designs.

Here, we introduce a universal diffractive waveguide framework that covers various functionalities, demonstrating low-loss guiding of desired modes as well as bent mode transmission (changing the direction of mode propagation), mode filtering, mode splitting, and mode-specific polarization-maintaining designs. These designs are cascadable to each other, forming any desired waveguide length and

[1]Electrical and Computer Engineering Department, University of California, Los Angeles, CA, USA. [2]Bioengineering Department, University of California, Los Angeles, CA, USA. [3]California NanoSystems Institute (CNSI), University of California, Los Angeles, CA, USA. [4]These authors contributed equally: Yuntian Wang, Yuhang Li. ✉e-mail: ozcan@ucla.edu

topology, and they employ successive diffractive layers optimized using deep learning[35–44] to minimize specific training functions tailored for desired applications. As a proof of concept, we demonstrated a diffractive waveguide design to match the performance of a traditional square dielectric waveguide, supporting the transmission of target spatial modes with high mode quality and low loss. A bent diffractive waveguide was also demonstrated to redirect the mode propagation direction, which is important for creating an arbitrary waveguide topology. The numerical testing results proved the feasibility of this framework, as the diffractive waveguide successfully transmitted target spatial modes with high coupling efficiency and low propagation loss through multiple cascades of the same diffractive unit. Additionally, we demonstrated programmable mode filtering diffractive waveguides that can selectively transmit or block specific optical modes, functioning as a high-pass or bandpass spatial mode filter for guided waves. Furthermore, a programmable mode splitting diffractive waveguide unit was also designed, which separated input modes into distinct output channels based on the mode order, illustrating low crosstalk between the output channels. The mode filtering and mode splitting functionalities were designed for both monochromatic and multi-wavelength light. Moreover, we demonstrated a programmable mode-specific polarization-maintaining diffractive waveguide, which maintained the polarization states of distinct selected spatial modes. Beyond numerical analyses, we also experimentally demonstrated 3D-printed mode filtering diffractive waveguides working at terahertz (THz) radiation for monochromatic as well as multi-wavelength operation, which allowed a certain set of spatial modes to pass while blocking the rest. The experimental results matched our simulations, confirming the feasibility of our diffractive waveguide framework.

In this work, we introduce and demonstrate a diffractive waveguide framework utilizing cascadable discrete diffractive layers that periodically modulate the phase structure of light without the need for any material dispersion engineering. These cascaded structured diffractive layers surrounded by air (or specific types of gas or liquid, if desired) give us new degrees of freedom to create and assemble task-specific waveguide topologies composed of repeating dielectric surfaces designed by deep learning. With minimal adjustments made according to the desired task, these diffractive waveguides can be optimized, offering a standardized approach to task-specific waveguide design. Additionally, these waveguides can be scaled to operate across different parts of the electromagnetic spectrum without the need to redesign their structures or material engineering, by simply scaling the diffractive feature size proportional to the wavelength of interest, forming a versatile waveguide platform, enhancing the flexibility of conventional designs[42]. Finally, diffractive waveguides can also control and multiplex the polarization states of guided modes by the integration of a 2D polarizer array (PA) within the diffractive unit design[38,45]. These diffractive waveguides will inspire further research and development, proving beneficial for a variety of applications in e.g., telecommunications, imaging, sensing, and spectroscopy.

## Results

### Diffractive waveguide design—multi-mode and single-mode designs

As a proof-of-concept demonstration, we designed a diffractive waveguide to match the performance of a square dielectric waveguide. Figure 1a illustrates the cross-section of a traditional square dielectric waveguide, consisting of core and cladding regions designed to support the propagation of certain spatial modes. The cross-sectional profile of the refractive index of this square dielectric waveguide is shown in Fig. 1b, where the refractive indices of the core and the cladding were selected as $n_{core} = 1.6$ and $n_{cladding} = 1.4$, respectively. Based on this configuration, the spatial modes $\{M_0, M_1, \ldots\}$ supported

by this square dielectric waveguide were calculated using an integrated full-vectorial finite-difference method-based mode solver[46,47]. In our analysis, the dielectric waveguide used for comparison was assumed to have an ideal refractive index profile, without any material defects and structural inhomogeneities. This assumption effectively eliminates various sources of losses for the standard dielectric waveguides considered in our comparison. Under these conditions, coupling and energy efficiencies are set to 100% for the traditional dielectric waveguides, serving as a reference baseline for evaluating the performance of our diffractive waveguide designs. Our diffractive waveguide, shown in Fig. 1c, was designed to support the propagation of the same target spatial modes and emulate the functionality of the traditional square dielectric waveguide under an illumination wavelength of $\lambda = 0.75$ mm. The cascadable unit architecture of our diffractive waveguide design consisted of two layers, spanning a total axial length of $53.3\lambda$, with each layer containing $200 \times 200$ trainable phase-only diffractive features, each with a size of ~$0.53\lambda$. During the training stage of our cascadable diffractive waveguide unit, the first 20 spatial modes $\{M_0, M_1, \ldots M_{19}\}$, calculated based on the assumed square dielectric waveguide cross-section (Fig. 1b), were fed as the complex-valued inputs to the diffractive waveguide input aperture. The phase modulation values of the diffractive features at each layer were iteratively optimized using stochastic gradient descent (SGD) to minimize a training loss function, which ensured the preservation of the targeted modes both in structural consistency (phase and amplitude) and output energy efficiency (see the "Methods" section). After its training, the resulting diffractive waveguide unit (Fig. 1d) could match the mode guiding performance of a traditional square dielectric waveguide (Fig. 1b). Figure 1e visualizes the output fields from both the traditional square dielectric waveguide and the corresponding diffractive waveguide unit, respectively. The optimized diffractive layers exhibit phase modulation patterns, structured at the diffraction limit of light, that sequentially relay the input light as it passes through the successive layers. The corresponding cross-sectional light field profile is also shown in Supplementary Fig. S1. The relative errors between the outputs of the two waveguides (dielectric vs. diffractive) are also displayed, showing that the diffractive waveguide can successfully transmit and maintain the complex-valued spatial modes supported in the traditional square dielectric waveguide with negligible error.

Besides testing the trained diffractive waveguide with the first 20 spatial modes $\{M_0, M_1, \ldots, M_{19}\}$ used during the training stage, we further evaluated the external generalization ability of the diffractive waveguide using higher-order modes $\{M_{20}, M_{21}, \ldots, M_{39}\}$ that were never used during training. The results of this comparative analysis for internal and external generalization ability of the diffractive waveguide are illustrated in Fig. 2a, showing a good match between the input and output mode profiles even for higher-order modes $\{M_{20}, M_{21}, \ldots, M_{39}\}$ never used in training. We also quantified the mode transmission quality of our diffractive waveguide using the coupling efficiency and energy efficiency metrics (see the "Methods" section). As shown in Fig. 2b, the coupling efficiency and the energy efficiency for internal generalization (green region) were consistently higher than 92.0% and 99.4%, respectively. For the higher-order spatial modes (pink region), the coupling efficiency and the energy efficiency remained above 84.0% and 99.0%, respectively. These results indicate that our diffractive waveguide not only learned to support the trained modes of interest ($\{M_0, M_1, \ldots, M_{19}\}$) but also generalized very well to guide the higher-order modes supported by the refractive index profile of a square dielectric waveguide—although they were never used in the training stage of the diffractive waveguide. To further investigate the coupling efficiency drop observed for higher-order modes, we analyzed the performance of diffractive waveguides trained on different spatial mode sets (see Supplementary Fig. S2). These results show that the coupling efficiency values are significantly improved when all the desired spatial modes of interest are known and accessible during training.

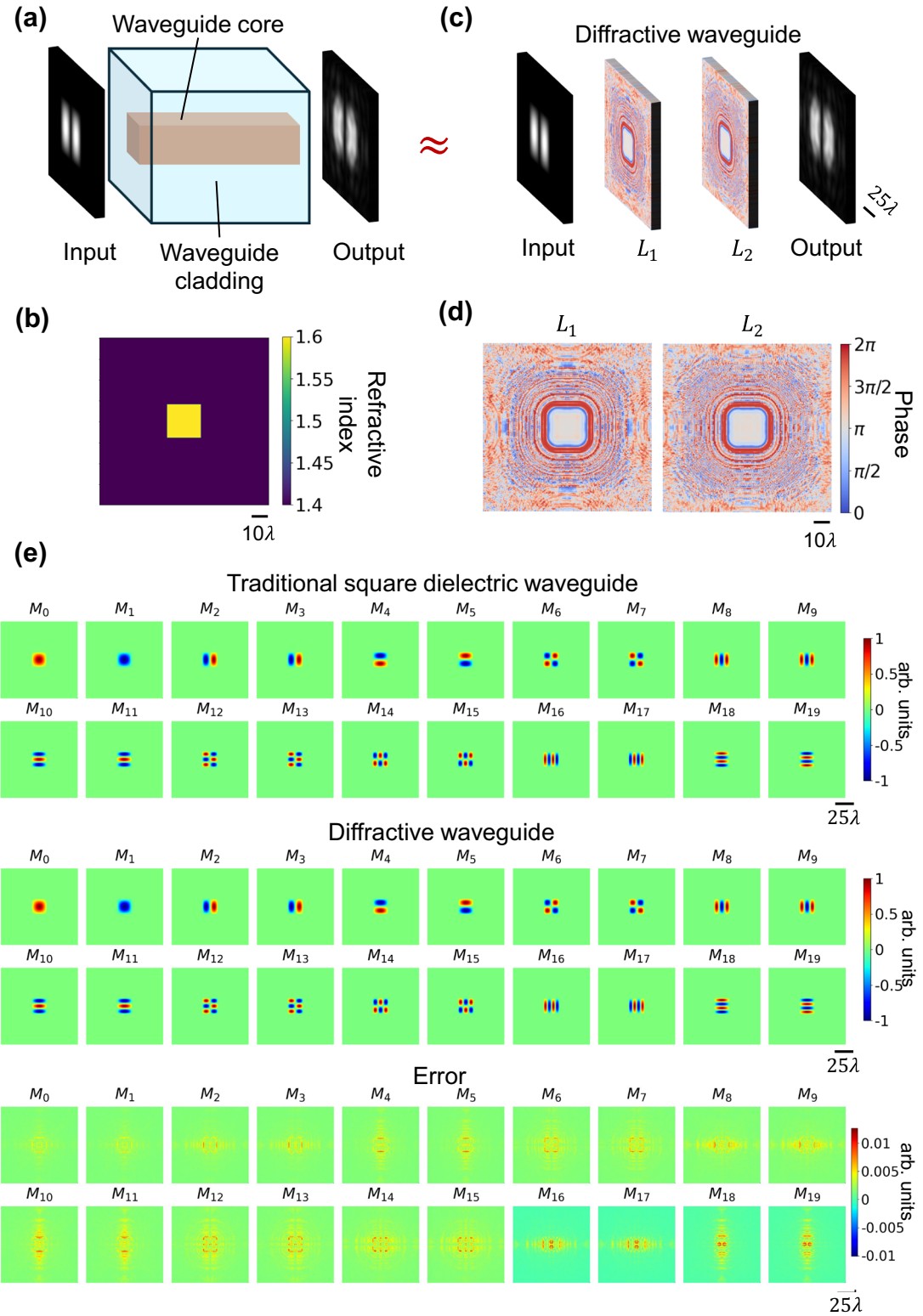

**Fig. 1 | Comparison between a traditional dielectric waveguide and a diffractive waveguide. a** The cross-section of a traditional square dielectric waveguide consisting of core and cladding dielectric regions, which transmit certain guided spatial modes with high fidelity. **b** The refractive index cross-sectional profile of the square dielectric waveguide. **c** Schematic of a cascadable diffractive waveguide unit designed to match the performance of the square dielectric waveguide. **d** Phase modulation patterns of the trained diffractive layers of the diffractive waveguide. **e** Output fields from the square dielectric waveguide and the diffractive waveguide are compared, with the relative errors displayed. The output fields of the traditional dielectric waveguide were calculated by a full-vectorial finite-difference method, serving as the ground truth for evaluating the output fields of the diffractive waveguide. The modes are denoted as $M_0 \sim M_{19}$.

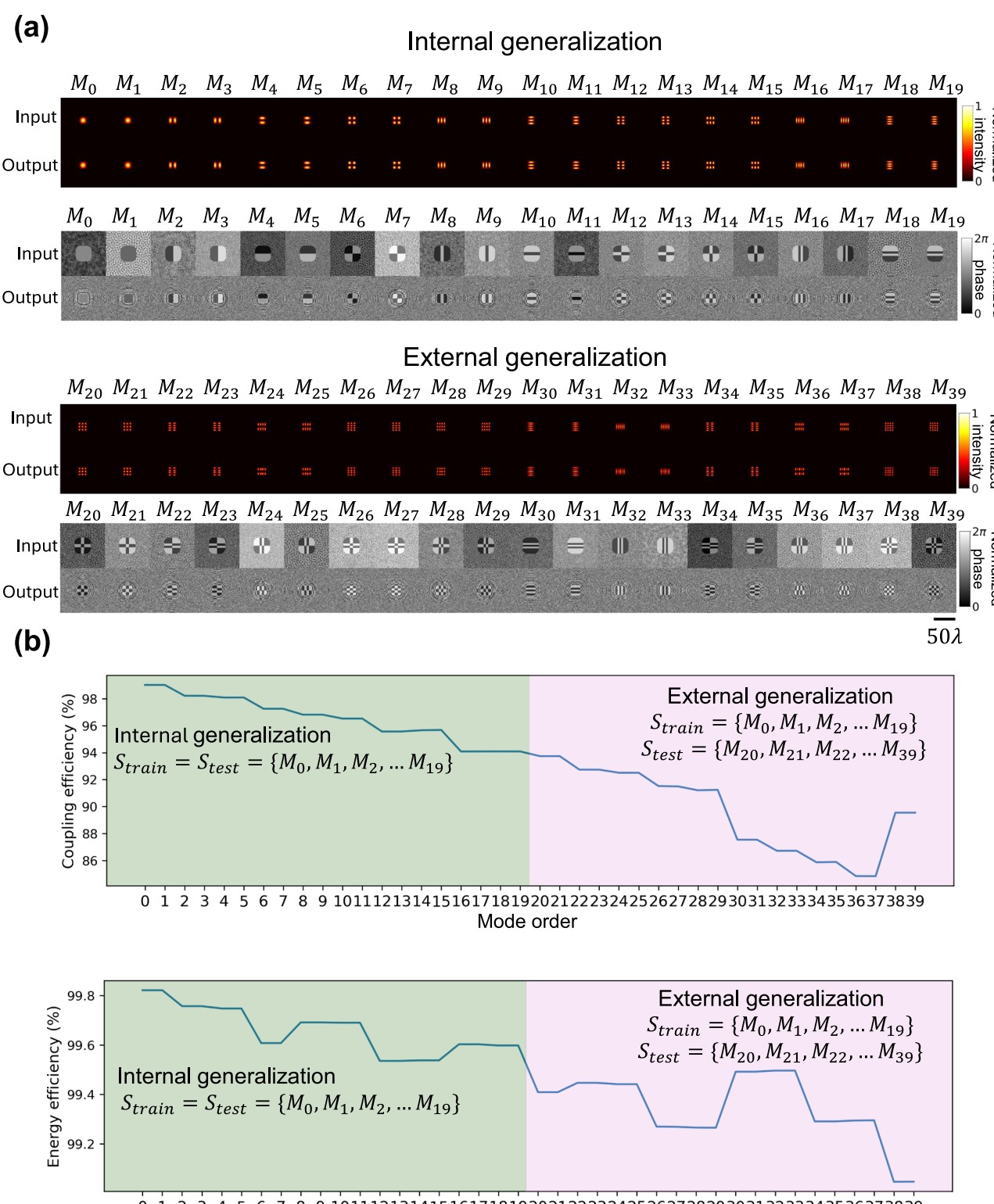

**Fig. 2 | Comparative analysis for internal and external generalization ability of the diffractive waveguide. a** Intensity and phase profiles of the input and output fields of the diffractive waveguide. Performance was evaluated using two sets of spatial modes: the internal generalization set $\{M_0, M_1, \ldots, M_{19}\}$, which was used during the training stage, and the external generalization set $\{M_{20}, M_{21}, \ldots, M_{39}\}$, which was never used during the training. **b** Coupling efficiency and energy efficiency of the transmitted spatial modes $\{M_0, M_1, \ldots M_{39}\}$. The internal generalization performance appears in the green region, and the external generalization performance appears in the pink region.

Additionally, to account for material absorption in practical implementations, we incorporated energy absorption into the training process, as shown in Supplementary Fig. S3. Compared to designs trained without considering material absorption, those optimized with absorption exhibited improved efficiency when tested with absorbing diffractive materials, demonstrating the benefit of accounting for such factors in the training loss function and optimization process.

Next we analyzed the cascading behavior of the optimized diffractive waveguide unit reported in Fig. 2. For this analysis, we cascaded $N = 10$ identical diffractive waveguide units to achieve a longer propagation distance, as depicted in Fig. 3a. In this cascaded design, any two successive diffractive waveguide units shared a common intermediate plane, i.e., the output field $o_i$ of the diffractive unit $i$ served as the input field for the subsequent diffractive unit $i + 1$. The intensity and phase of the optical field at the input plane, intermediate plane, and output plane are visualized in Fig. 3b, where the intensity patterns were normalized and the phase patterns were adjusted by subtracting a uniform constant phase shift. It can be seen that the spatial modes of the cascaded diffractive waveguide were maintained very well after wave propagation through 10 cascaded diffractive units. Furthermore, the coupling efficiency and the energy efficiency of different modes at each intermediate plane are reported in Fig. 3c, d. These results revealed that: (i) with more diffractive units cascaded, the coupling efficiency of each mode relatively decreased but remained above 91% for all the modes; (ii) the energy efficiency of each mode dropped monotonically with the increase of cascaded diffractive waveguide units but remained over 70% after 10 units; and (iii) higher-order modes experienced a more significant drop in the coupling efficiency and the energy efficiency, which may be due to the higher frequency components contained in these modes, requiring more trainable degrees of freedom in the design of the diffractive waveguide. These results suggest that our diffractive waveguide not only supports the desired modes of interest but also functions as a modular, Lego-like component to be cascaded without further training. This plug-and-play-ready modularity is important for designing solutions to satisfy the needs of various waveguide applications. If needed, however, the overall performance of a cascaded diffractive waveguide of any arbitrary length ($N \gg 1$) can be further improved by using transfer learning and training the cascaded diffractive structure in an end-to-end manner for a given axial length of interest; this would further improve the performance of the diffractive waveguide for a given/desired topology. As reported in Supplementary Fig. S4, when a specific topology is required, end-to-end optimization delivers superior performance in both the coupling efficiency and the energy efficiency compared to the unit-level optimization. While the end-to-end design approach excels in these performance metrics, unit-level optimization is better suited for more general scenarios regardless of the number of cascading units. Moreover, unit-level optimization requires only a one-shot training procedure, unlike the end-to-end approach, which needs to be designed on a case-by-case basis, different for each application.

In addition to the multi-mode waveguide designs reported earlier, we also demonstrated single-mode diffractive waveguides. To match the mode profile of a commonly used optical fiber (SMF-28)[48] in telecommunication systems, we designed a diffractive waveguide that operates at 1550 nm; see Supplementary Fig. S5. In this infrared waveguide design reported in Supplementary Fig. S5, the feature size of the diffractive layers was scaled down for operation at 800 nm, while the axial distance between successive diffractive layers was selected as 100 µm, corresponding to the common thickness of various glass substrates that can potentially be stacked together. The single-mode transmission behavior and the cascading capability of this diffractive waveguide design were successfully demonstrated and quantified in Supplementary Fig. S5. To explore the feasibility of implementations in the near-infrared part of the spectrum, we conducted simulations at 1550 nm

wavelength for approximating the waveguiding behavior of SMF-28 fiber[48], which is a single-mode optical fiber that is widely used in telecommunications. As shown in Supplementary Fig. S6, >95% coupling efficiency and >95% energy efficiency can be achieved with 4-bit depth diffractive layers ($K = 2$), with each layer having a lateral resolution of 1.55 µm. These fabrication specifications in terms of phase bit depth and lateral resolution are compatible with commercial two-photon polymerization-based 3D nanoprinters or lithography-based nanofabrication technologies. Especially wafer-scale high-throughput fabrication of such diffractive designs using high-purity fused silica that has an ultra-low loss and high thermal stability, as demonstrated in recent work[49], holds promise for mass-scale fabrication of cascadable diffractive waveguides operating in the near-infrared spectrum[42,50]. As another example of a single-mode diffractive waveguide design with a tighter mode diameter, we selected 532 nm as our operation wavelength, where the resulting diffractive design confined the full width at half maximum of the spatial mode supported by the diffractive waveguide to $\sim 6.4\lambda = 3.4$ µm, as shown in Supplementary Fig. S7. These results illustrate the diffractive waveguide's capability to confine single-mode light within a tight modal profile, comparable to traditional planar, strip-on-chip waveguides, underscoring its potential for broader applications.

Although the diffractive waveguides demonstrated so far exhibit increased total energy loss with longer propagation distances, it is crucial to highlight that their versatile functionalities—such as mode filtering and mode splitting— provide some unique features. Unlike optical fibers, which are optimized for long-distance transmission, one of the key objectives of these diffractive waveguides is to provide greater design flexibility for short-range transmission and light processing applications, such as, e.g., on-chip optical interconnects, multiplexed routing, and distributed sensors. Diffractive waveguides and the design framework behind them expand the range of tools available for creating customized waveguide designs tailored for various spatial, spectral, and/or polarization-based light processing, multiplexing, and routing needs.

## Bent diffractive waveguide design

One of the important functions of waveguides is to efficiently guide and redirect light. To address this need, we designed a bent diffractive waveguide to redirect the guided modes of light and provide an interface that can be directly cascaded with the linear diffractive waveguide design described in the previous sub-sections. Figure 4a illustrates the structure of a bent diffractive waveguide with four layers, in which the light propagation direction was rotated by 45° from the input to the output plane. Here, the input plane is parallel to the first diffractive layer, and the output plane is parallel to the last diffractive layer. Starting from the second diffractive layer of the bent waveguide design, a 15° clockwise rotation was performed relative to the normal direction of the previous plane, culminating in 45° rotation in total. The tilted angular spectrum method[51] was used to simulate light propagation between the adjacent diffractive layers, and these diffractive layers were trained to minimize the structural difference between the input optical modes and the output optical fields based on the coupling efficiency metric, while also maximizing the output energy efficiency (see the "Methods" section for details). Once the training converged, the trained diffractive layers, as shown in Fig. 4b, were numerically tested with the optical modes of the same dielectric square waveguide, including both $\{M_0 \sim M_{19}\}$ that were used in training (internal generalization) and the higher-order modes $\{M_{20} \sim M_{39}\}$ that were never used in training (external generalization). The intensity and phase profiles of the input and the transmitted optical fields at the output are visualized in Fig. 4c, showing that all the modes $\{M_0 \sim M_{39}\}$ could be maintained after propagating through the bent diffractive waveguide. Figure 4d

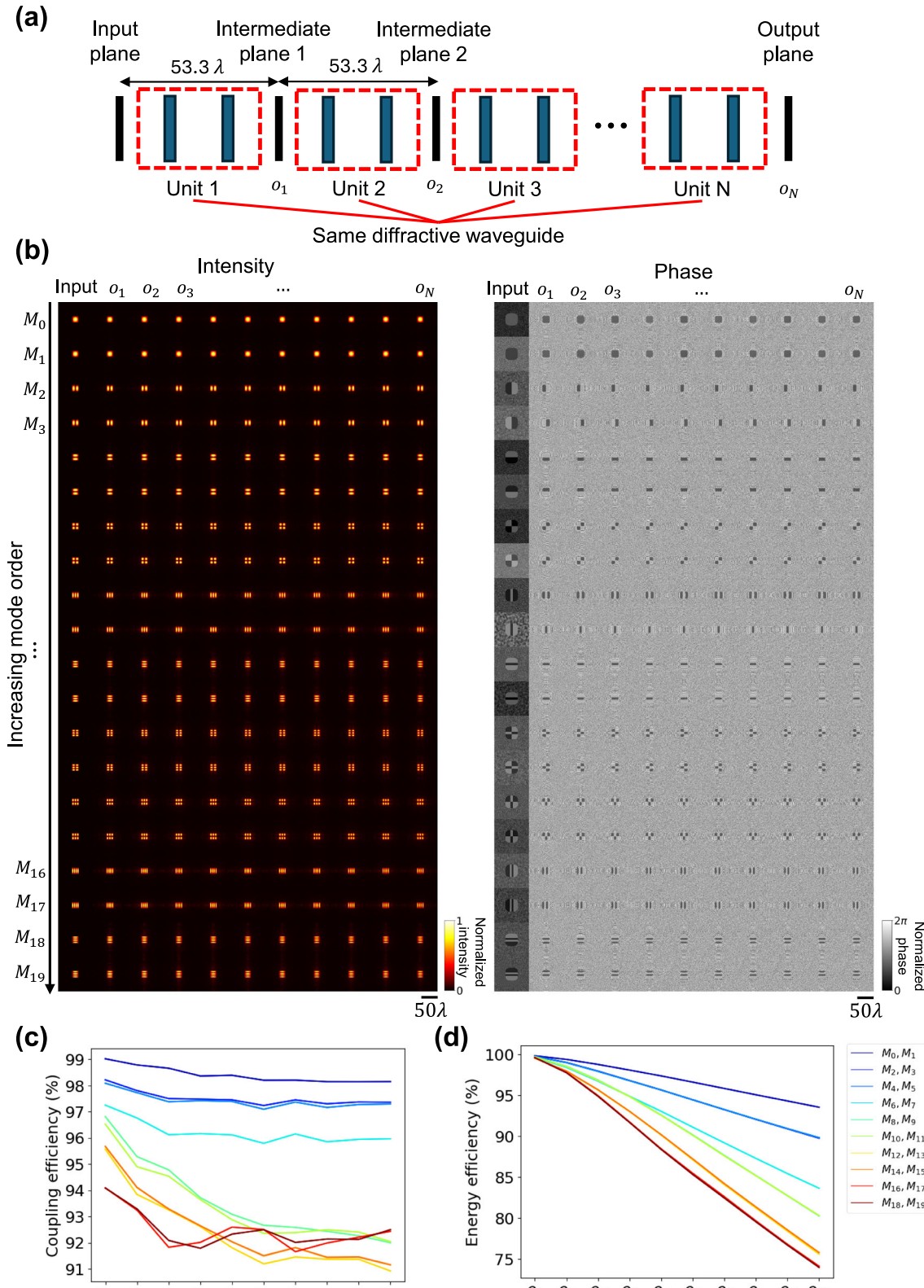

**Fig. 3 | Testing results of a cascaded diffractive waveguide for mode transmission. a** Schematic of a cascaded diffractive waveguide with *N* identical diffractive units. **b** Intensity and phase profiles of optical fields at the input, intermediate, and output planes. **c** Coupling efficiency and **d** energy efficiency of different transmitted modes $\{M_0, M_1, \ldots, M_{19}\}$ at intermediate planes and the output plane. The curves in different colors correspond to different mode orders shown in the legend.

reports the coupling efficiency and the energy efficiency of these spatial modes after transmitting through the bent diffractive waveguide. The coupling efficiency for internal generalization remained >90% while the energy efficiency was greater than 60%. Although the higher-order modes $\{M_{20} \sim M_{39}\}$ were never used during the training process, the bent diffractive waveguide achieved >75% coupling efficiency and >65% energy efficiency for these modes, indicating the success of the bent diffractive waveguide design.

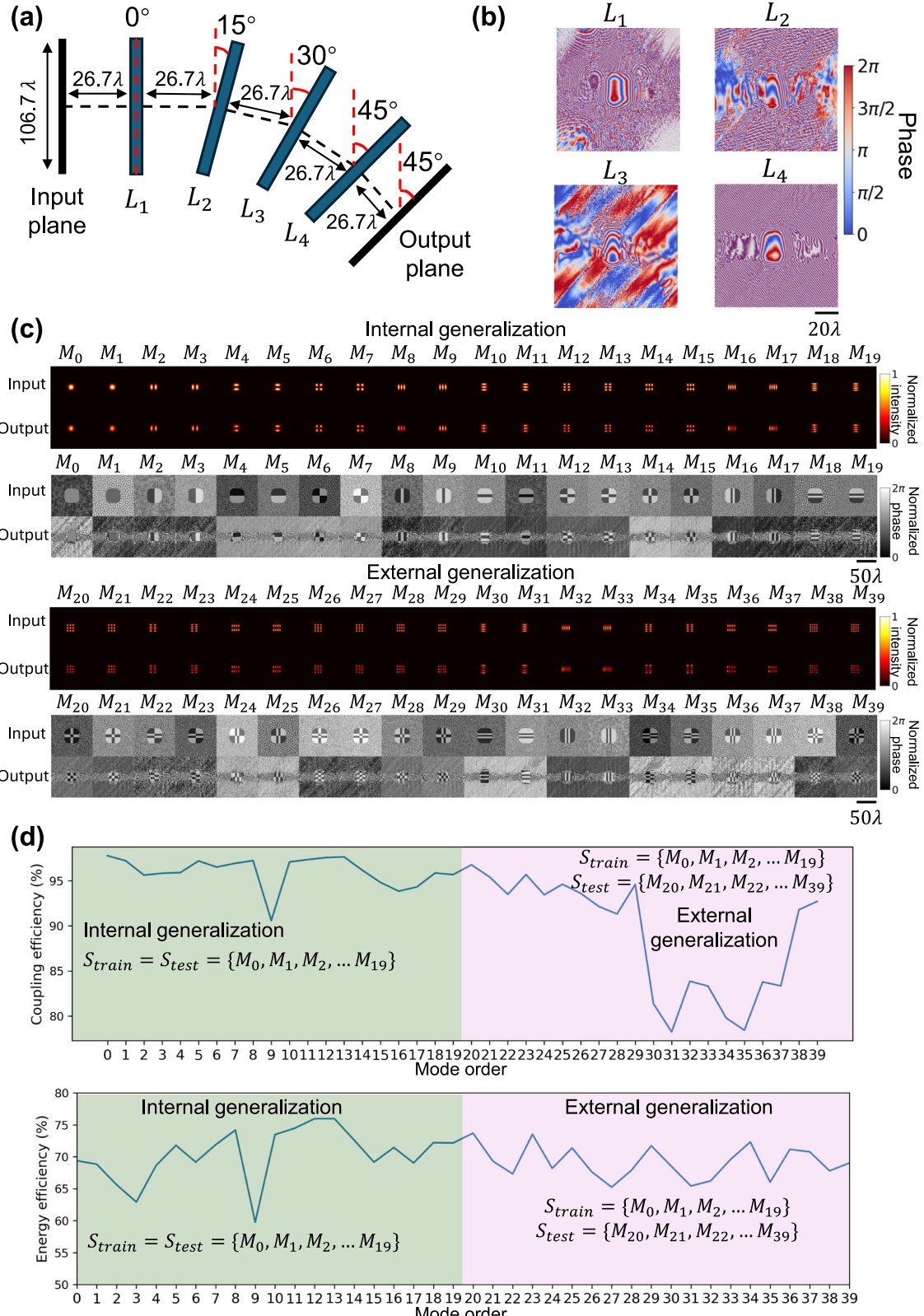

**Fig. 4 | The structure and testing results of a bent diffractive waveguide.**
**a** Illustration of a bent diffractive waveguide. **b** Phase modulation patterns of the converged diffractive layers of the bent diffractive waveguide. **c** Intensity and phase profiles of the input and output fields of the bent diffractive waveguide. Performance was evaluated using two sets of spatial modes: an internal set $\{M_0, M_1, \ldots, M_{19}\}$, which was used during the training, and an external set $\{M_{20}, M_{21}, \ldots, M_{39}\}$, which was never used during the training. Each row of output images is separately normalized. **d** Coupling efficiency and energy efficiency of the transmitted modes $\{M_0, M_1, \ldots, M_{39}\}$ at the output plane. The internal generalization performance appears in the green region, and the external generalization performance appears in the pink region.

To further investigate the impact of the number of diffractive layers and the rotation angle between two successive layers on the performance of the tilted diffractive waveguide, various designs with different numbers of layers and bending angles were explored. As shown in Supplementary Fig. S8, increasing the number of diffractive layers for a fixed total bending angle improved waveguide performance due to the greater design flexibility and larger degrees of freedom available, leading to better coupling efficiencies. Additionally, the influence of different total bending angles while maintaining a constant turn step (11.25°) and a constant axial distance between two successive layers (26.67$\lambda$) was also examined; see Supplementary Fig. S9. While the coupling efficiency remained consistent among different conditions, the energy efficiency slightly declined with increasing propagation distance and bending angle. Notably, there is no single optimal solution as each diffractive waveguide design can be optimized based on specific application requirements. These analyses underscore the flexibility of diffractive waveguide designs, enabling tailored solutions for various desired performance specifications.

## Mode filtering diffractive waveguide design

Spatial mode filters are foundational in controlling the transmission and rejection of specific guided modes, essential for various optical and photonic applications, including, e.g., optical communications, microscopy, and laser engineering[20–22]. However, these filters are often bulky, costly, and difficult to integrate into compact systems[52,53]. Additionally, traditional waveguide filters struggle to precisely control the spatial mode selections based on the mode number and/or frequency. Our diffractive waveguides can be designed by appropriately tailoring the training loss function for a specific desired task, providing a versatile approach to cover various applications. To showcase this, we designed mode filtering diffractive waveguides capable of selectively passing or rejecting certain spatial modes of interest. These mode filtering diffractive waveguides were precisely optimized to transmit desired modes while effectively blocking unwanted guided modes. We demonstrated this capability through two different designs: a high-pass mode filtering waveguide and a bandpass mode filtering waveguide. The high-pass filtering waveguide differentiated between a target set of modes to be passed/transmitted, i.e., $S_t = \{M_{12}, M_{13}, M_{14}, \ldots M_{19}\}$ and a rejection mode set, i.e., $S_r = \{M_0, M_1, M_2, \ldots, M_{11}\}$; in this first design, we preserved the energy efficiency and mode patterns for the target set $S_t$ while filtering out the guided modes of the rejection set $S_r$—see Supplementary Fig. S10a. In the second mode filtering diffractive waveguide design, we aimed at bandpass filtering, where the transmission set was defined as $S_t = \{M_0, M_1\} \cup \{M_6 \sim M_{11}\} \cup \{M_{16} \sim M_{19}\}$ and the rejection set was defined as $S_r = \{M_2 \sim M_5\} \cup \{M_{12} \sim M_{15}\}$.

To design these mode filtering diffractive waveguides, four hyperparameters were defined: $T_{CE,u}$, $T_{CE,l}$, $T_{E,u}$ and $T_{E,l}$, which set the upper ($u$) and lower ($l$) thresholds of the coupling efficiency ($T_{CE,u}$, $T_{CE,l}$) and the energy efficiency ($T_{E,u}$ and $T_{E,l}$), respectively. By adjusting these threshold values and following a similar training process as described in the previous sub-sections, mode filtering diffractive waveguide designs with different passband and rejection band metrics could be achieved (see the "Methods" section for details). Supplementary Fig. S10b reports the input and output patterns of the high-pass mode filtering diffractive waveguide along with the resulting coupling efficiency and energy efficiency values. This high-pass mode filtering diffractive waveguide design successfully rejected, as expected/desired, unwanted lower-order modes with an energy efficiency < 10% and a coupling efficiency ~ 3%, and allowed the desired spatial modes with > 50% energy efficiency and > 90% coupling efficiency. Similarly, the results of the bandpass mode filtering diffractive waveguide design are shown in Supplementary Fig. S10c, where $S_r = \{M_2 \sim M_5\} \cup \{M_{12} \sim M_{15}\}$ were rejected with low coupling efficiency and energy efficiency values, and

$S_t = \{M_0, M_1\} \cup \{M_6 \sim M_{11}\} \cup \{M_{16} \sim M_{19}\}$ were transmitted through the diffractive waveguide with high quality. These results underscore the effectiveness of our mode filtering diffractive waveguide designs. By tuning the thresholds ($T_{CE,u}$, $T_{CE,l}$, $T_{E,u}$ and $T_{E,l}$) during the training stage, one can further design diffractive waveguides with different sets of guided modes that are selectively transmitted and rejected, streamlining the design of mode filtering diffractive waveguides for various applications.

To further demonstrate the unique capabilities of these mode filtering diffractive waveguide designs, a multi-wavelength mode filter was developed, as shown in Fig. 5a. This diffractive waveguide was optimized to selectively transmit different sets of spatial modes at certain illumination wavelengths. Specifically, the spatial modes in $S_{\lambda_1} = \{M_0, M_1\}$, $S_{\lambda_2} = \{M_6 \sim M_{11}\}$ and $S_{\lambda_3} = \{S_{16} \sim S_{19}\}$ can only transmit at the corresponding illumination wavelengths $\lambda_1 = 0.7$ mm, $\lambda_2 = 0.75$ mm, and $\lambda_3 = 0.8$ mm, respectively. All other modes were filtered out through this multi-wavelength mode filtering diffractive waveguide. Following a similar training strategy used for its monochromatic counterpart, the multi-wavelength design was effectively optimized to select distinct sets of modes at different illumination wavelengths. Figure 5b illustrates the input and output patterns across different wavelengths. The resulting coupling efficiency and the energy efficiency values, shown in Fig. 5c, further highlight the success of the multi-wavelength filter function, where the desired, transmitted modes maintain high quality, and the rejected modes exhibit low coupling and energy efficiencies within the targeted wavelengths. By adjusting the loss function during the training phase, diffractive waveguides with tailored functionalities for more advanced mode filtering across specific mode orders and wavelengths can be achieved.

## Mode splitting diffractive waveguide design

Different from the monochromatic and multicolor mode filtering diffractive waveguide designs reported in earlier sub-sections, here we consider the design of a diffractive waveguide that performs mode splitting from the input field of view (FOV) to separate regions in the output FOV (detailed in the "Methods" section). As illustrated in Supplementary Fig. 11a, the input modes $S_{in} = \{M_0, M_1, \ldots, M_{19}\}$ entering the input FOV were separated into two output channels based on their spatial mode order: lower-order modes in $S_{CH1} = \{M_0, M_1, \ldots, M_{11}\}$ were directed to Channel 1, while the higher-order modes in $S_{CH2} = \{M_{12}, M_{13}, \ldots, M_{19}\}$ were directed to Channel 2. To optimize this mode splitting diffractive waveguide, we superimposed spatial modes of varying orders and minimized the mean square error (MSE) between the presumed weights for the input mode combination and the decomposed weights derived from the target spatial mode at each output channel. The resulting trained diffractive layers are reported in Supplementary Fig. 11b. To blindly test this mode splitting diffractive waveguide design, we sequentially input different modes at the input FOV; see Supplementary Fig. 11c. The mode splitting diffractive waveguide successfully guided all the modes in $S_{CH1} = \{M_0, M_1, \ldots, M_{11}\}$ into Channel 1, while the modes in the second set $S_{CH2} = \{M_{12}, M_{13}, \ldots, M_{19}\}$ were directed to Channel 2—as targeted/desired. Furthermore, the coupling efficiency and the energy efficiency of the two channels for all the spatial modes of interest were quantified in Supplementary Fig. 11d, showing a good mode splitting performance, also suppressing the energy of the unwanted modes in each output channel. We also trained mode splitting diffractive waveguides with different numbers ($K$) of diffractive layers in each unit, revealing that the energy efficiency for both output channels increased with more diffractive layers, showcasing the advantages of deeper diffractive designs with more degrees of freedom (see Supplementary Fig. 11d). Although the coupling efficiency of the filtered modes in CH1 was appreciable, the corresponding energy efficiency was notably diminished—as desired, indicating that only a minimal amount of energy was coupled into these specific spatial modes. In various

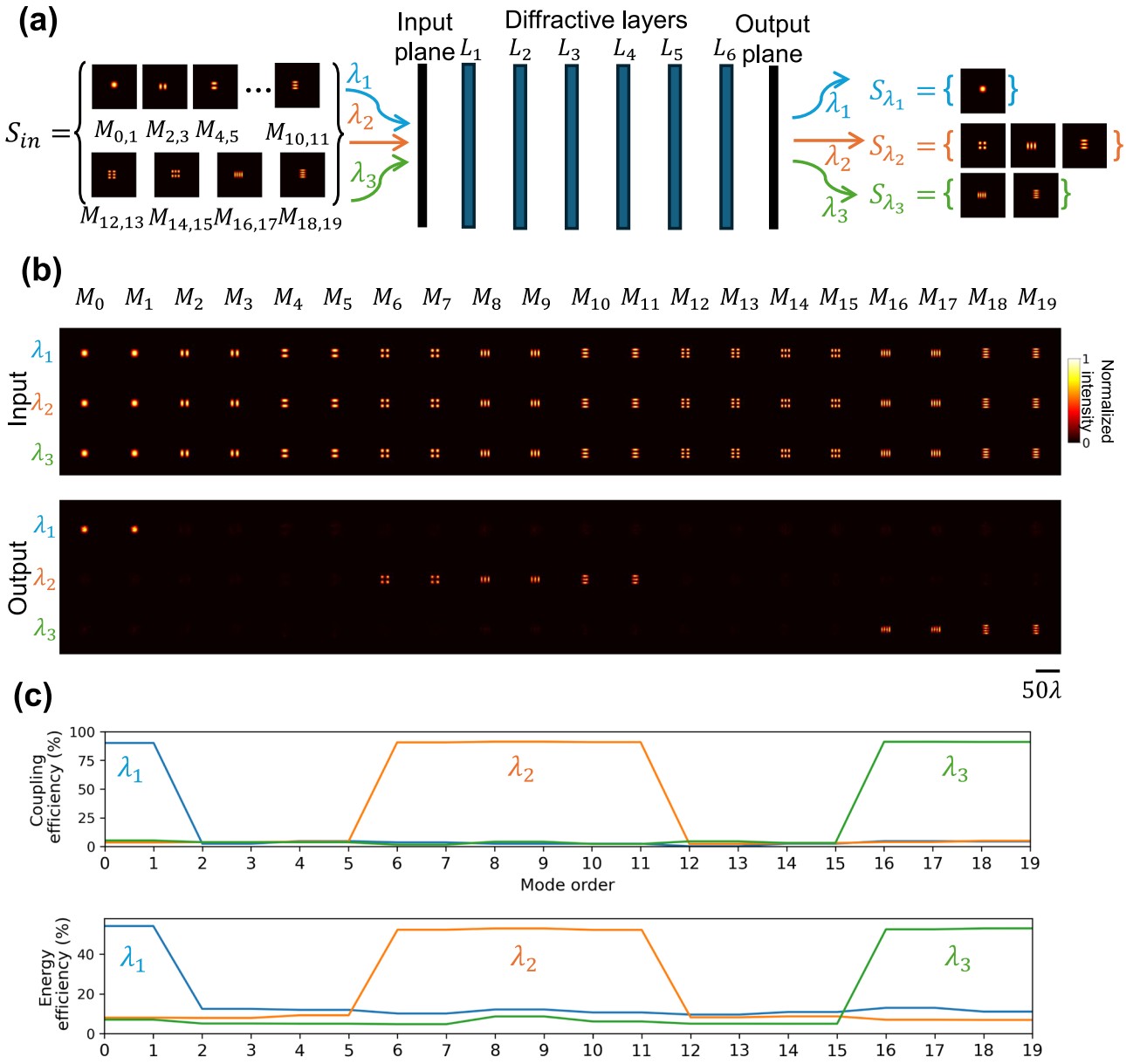

**Fig. 5 | Testing results of a multi-wavelength mode filtering diffractive waveguide. a** Schematic of a multi-wavelength mode filtering diffractive waveguide designed to pass different sets of modes at different illumination wavelengths. Spatial modes in $S_{\lambda_i}$ can only be transmitted at the corresponding illumination wavelength $\lambda_i$ ($i = 1, 2, 3$). **b** Intensity profiles of the input and output fields of the multi-wavelength mode filtering diffractive waveguide. **c** Coupling efficiency and energy efficiency of all the input spatial modes at different illumination wavelengths.

applications such as wavelength-division multiplexing (WDM) and mode-division multiplexing (MDM), the energy efficiency serves as a more critical parameter for evaluating performance. We also observed that the coupling efficiency of suppressed modes in CH1 and CH2 differed, which can be attributed to the hyperparameters $\alpha$ and $\beta$, which were adjusted to balance the coupling efficiency and energy efficiency. In this work, the energy efficiency was prioritized, as it is essential for the optimal performance of the mode filtering diffractive waveguide. To assess the influence of these hyperparameters, $\alpha$ in Eq. 19 was varied from 10 to 50 in increments of 10. The results, presented in Supplementary Fig. S12, revealed that increasing $\alpha$ led to a further reduction in the coupling efficiency of the filtered modes—as desired.

In a more general scenario, spatial modes with multiple wavelengths within a single input aperture can also be demultiplexed into different channels. To demonstrate this capability, we report a multi-

wavelength mode splitting diffractive waveguide that can split the input aperture into $2 \times 2$ outputs based on the order of the spatial modes and illumination wavelengths. As illustrated in Fig. 6a, the input modes in two different illumination wavelengths entering the input aperture were separated into four output channels based on their wavelength and spatial mode order: at illumination wavelength $\lambda_1 = 0.7 \, \text{mm}$, modes in $S_{CH1} = \{M_0 \sim M_{11}\}$ and $S_{CH2} = \{M_{12} \sim M_{19}\}$ were directed into Channel 1 and Channel 2, respectively; at illumination wavelength $\lambda_2 = 0.8 \, \text{mm}$, modes in $S_{CH3} = \{M_0 \sim M_7\}$ and $S_{CH4} = \{M_8 \sim M_{19}\}$ were directed into Channel 3 and Channel 4, respectively. The trained model was blindly tested with different orders of optical modes at these two illumination wavelengths individually, shown in Fig. 6b. At $\lambda_1$ illumination, the multi-wavelength mode splitting diffractive waveguide successfully guided all the modes in $S_{CH1}$ into Channel 1 and the other modes in $S_{CH2}$ into Channel 2 while exhibiting no apparent crosstalk with Channels 3 and 4. Similarly,

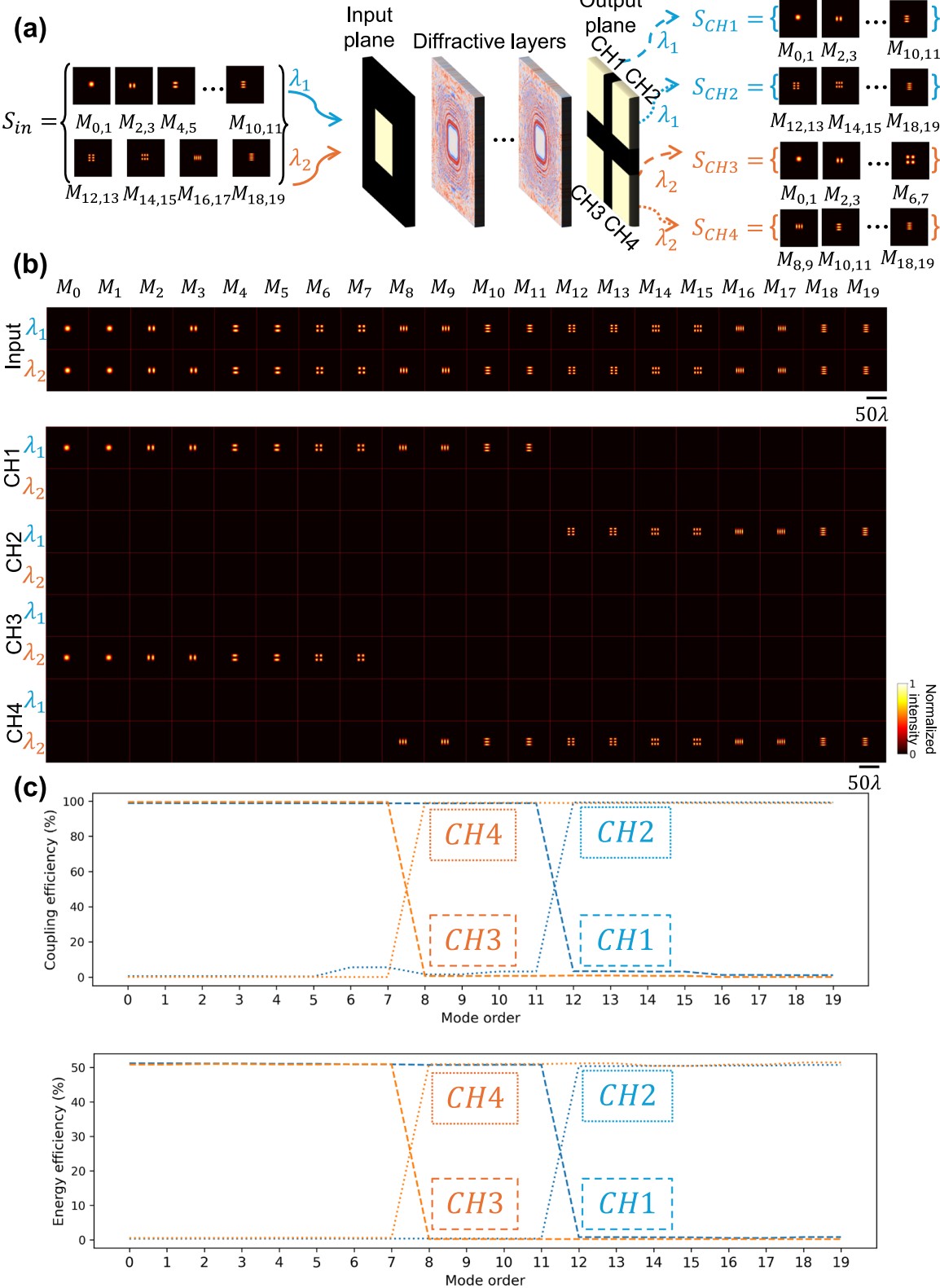

**Fig. 6 | Testing results of a multi-wavelength mode splitting diffractive waveguide. a** Schematic of a multi-wavelength mode splitting diffractive waveguide, designed to split modes from the input aperture to separate regions in the output aperture, depending on both the order of the spatial modes and the wavelengths. **b** Blind testing results: the intensity profiles of the input fields and the output fields at the four output channels of the multi-wavelength mode splitting diffractive waveguide. Each row of output images was separately normalized. **c** Coupling efficiency and energy efficiency of the four output channels for the transmitted modes $\{M_0, M_1, \ldots, M_{19}\}$ at the output plane.

Channels 3 and 4 were able to receive their desired set of modes at the illumination wavelength $\lambda_2$ with no evident leakage into Channels 1 and 2. Furthermore, we quantified the coupling efficiency and the energy efficiency of these four output channels for all spatial modes of interest in Fig. 6c, showing a good mode splitting performance and suppression of unwanted modes in each output channel.

### Mode-specific polarization-maintaining diffractive waveguide

We also designed mode-specific polarization-maintaining diffractive waveguides using a combination of optimized diffractive layers and PAs as illustrated in Fig. 7a. The PA consisted of multiple linear polarizer units with four different polarization directions: 0°, 45°, 90°, and 135°, which repeat periodically over space—i.e., non-trainable. The mode-specific polarization-maintaining diffractive waveguide was designed to maintain the $\hat{x}$ polarization state of the spatial modes in $S_{\hat{x}} = \{M_0, M_1, M_4, M_5, M_8, M_9, M_{12}, M_{13}, M_{16}, M_{17}\}$ while blocking all the spatial modes in $S_{\hat{y}} = \{M_2, M_3, M_6, M_7, M_{10}, M_{11}, M_{14}, M_{15}, M_{18}, M_{19}\}$. Symmetrically, it maintained the $\hat{y}$ polarization state of the spatial modes in $S_{\hat{y}}$ while blocking all the spatial modes in $S_{\hat{x}}$, as depicted in Fig. 7b. Figure 7c shows the coupling efficiency and the energy efficiency for the modes with the correct set of mode order and polarization state, while all the other undesired combinations of modes and polarization states were effectively filtered out. These results indicate that our mode-specific polarization-maintaining diffractive waveguide can be designed to maintain different sets of modes at different polarization states. The cascading behavior of this mode-specific polarization-maintaining diffractive waveguide was also investigated, reported in Supplementary Fig. 13. In this analysis, we cascaded $N = 5$ identical polarization-maintaining diffractive waveguide units to achieve a longer propagation distance, following a similar approach to the cascading diffractive waveguide design reported in Fig. 3. The intensity of the optical field at the input plane, intermediate planes, and output plane are visualized in Supplementary Fig. 14a. Moreover, the coupling efficiency and the energy efficiency of different guided modes, across all combinations of input and output polarization states at each intermediate plane, are presented in Supplementary Fig. 14b, which further illustrate the plug-and-play-ready feature of each polarization-maintaining diffractive waveguide unit, cascaded to each other, one following another.

### Experimental demonstration of mode filtering diffractive waveguides

We experimentally demonstrated the proof of concept of our mode filtering diffractive waveguide design using terahertz radiation. For this, we trained a three-layer mode filtering diffractive waveguide to selectively allow $\{M_6, M_7\}$ to pass through while rejecting the other spatial modes, as shown in Fig. 8a. Accordingly, we adjusted the hyperparameters $(T_{CE,u}, T_{E,u}, T_{CE,l}, T_{E,l}) = (0.85, 0.5, 0.05, 0.05)$ (detailed in the "Methods" section). Numerical analysis of the coupling efficiency and the energy efficiency of the resulting design confirmed that the mode filtering diffractive waveguide was successfully trained to selectively pass the specific spatial modes $\{M_6, M_7\}$ as shown in Fig. 9a. After the training stage, the resulting diffractive layers (Fig. 8b) were 3D printed, physically forming a mode filtering diffractive waveguide. This assembled diffractive waveguide was then experimentally tested under a THz scanning system shown in Fig. 8c, d.

For the experimental testing of our 3D-printed mode filtering diffractive waveguide, we fabricated four amplitude-only apertures and six phase-only apertures, which were used to generate structured optical fields, containing certain spatial modes at the input plane, as illustrated in the first and the second rows of Fig. 9b (see the "Methods" section for details). For these 10 input test apertures, all the amplitude-only apertures and the first three phase-only apertures could excite the input modes $\{M_6, M_7\}$; the input fields produced by the remaining three phase-only input apertures did not include these target spatial

modes. The output FOV of the mode filtering diffractive waveguide was scanned for each one of these 10 test input apertures using a THz detector, forming the output images displayed in the fourth row of Fig. 9b. These experimental results showed a decent agreement with the numerical simulation results and revealed that the 3D fabricated mode filtering diffractive waveguide effectively filtered out undesired spatial modes while permitting the transmission of the target spatial modes $\{M_6, M_7\}$—as desired in its design. The relatively small mismatch between the numerical simulation and experimental results may originate from the imperfections of the 3D fabrication and the misalignments between the diffractive layers, which can be mitigated by taking these factors into account during the training phase through, e.g., a numerical vaccination process[54].

In addition to our monochromatic experimental results, we also designed a multi-wavelength mode filtering diffractive waveguide to pass different sets of modes at different illumination wavelengths, as depicted in Fig. 10a. We followed the same process mentioned in the monochrome design while involving two illumination wavelengths, $\lambda_1 = 0.7$ mm and $\lambda_2 = 0.8$ mm. Once the model converged, the diffractive layers were 3D printed and aligned, as shown in Fig. 10b. We numerically calculated the coupling efficiency and the energy efficiency at the output plane for spatial modes at different illumination wavelengths, as shown in Fig. 10c. This multi-wavelength mode filtering diffractive waveguide design successfully passed $\{M_2, M_3\}$ at the illumination wavelength $\lambda_1 = 0.7$ mm and passed $\{M_{12}, M_{13}\}$ at the illumination wavelength $\lambda_2 = 0.8$ mm, while rejecting all other modes at both wavelengths. The fabricated multi-wavelength mode filtering diffractive waveguide was tested experimentally using four different phase-only apertures. These 3D-printed phase apertures, as illustrated in the first and second columns of Fig. 10d, were used to generate specific structured optical fields at different illumination wavelengths, shown in the third and fourth columns of Fig. 10d, which contained different ratios of optical modes. The first phase aperture could excite the corresponding target modes for both illumination wavelengths, the second and third apertures could individually excite the target modes at different wavelengths, while the optical field produced by the fourth phase aperture did not include any target modes. The outputs of this 3D fabricated multi-wavelength mode filtering diffractive waveguide were numerically simulated and experimentally measured using the same setup. Cross-correlation operation was used to evaluate the similarity between the experimental results and the simulated/numerical results (see the "Methods" section), which showed a good agreement between the two, experimentally demonstrating the effective mode filtering functionality of the 3D-printed multi-wavelength mode filtering waveguide.

## Discussion

In this work, we introduced various cascadable diffractive waveguide designs for different functionalities, including bent waveguides, mode filtering, and mode splitting, as well as mode-specific polarization-maintaining designs. Unlike traditional dielectric waveguides that rely on refractive index profiles of materials, our diffractive waveguides use cascaded phase modulation through spatially optimized diffractive layers. This simplifies the design of task-specific waveguides, where the training loss function of a cascadable diffractive waveguide unit can be optimized for different goals covering various spectral, spatial, and polarization features of interest, without the need for material dispersion engineering.

In contrast to earlier diffractive optics and metasurface-based studies that generally focus on isolated or single-function devices, our work presents a unified, diffractive waveguide platform that supports a broad range of integrated functionalities. Notably, the presented diffractive waveguide platform enables spatial and spectral mode filtering and mode splitting designs, as well as mode-specific polarization maintenance, offering a level of control that is

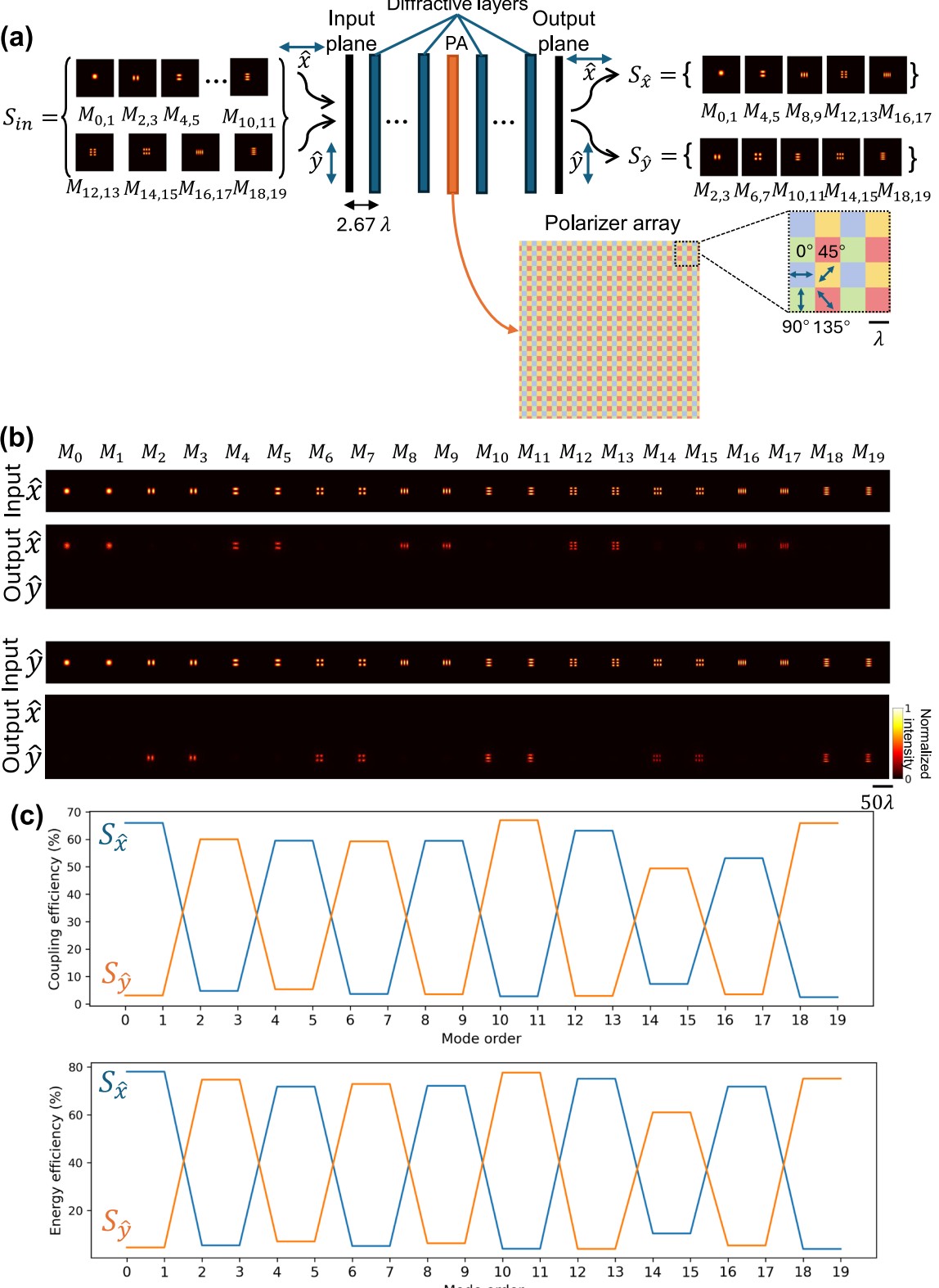

**Fig. 7 | The structure and testing results of a mode-specific polarization-maintaining diffractive waveguide. a** Illustration of a mode-specific polarization-maintaining diffractive waveguide. Spatial modes in each set can only be transmitted at a specific input polarization direction. The polarization array consists of multiple linear polarizer units with four different polarization directions: 0°, 45°, 90°, and 135°, which repeat periodically. **b** Blind testing results: the intensity profiles of the input fields and the output fields at two orthogonal linear polarization states ($\hat{x}$ and $\hat{y}$). Output images from the same input polarization direction were normalized together. **c** Coupling efficiency and energy efficiency of the transmitted modes at different combinations of input and output polarization directions.

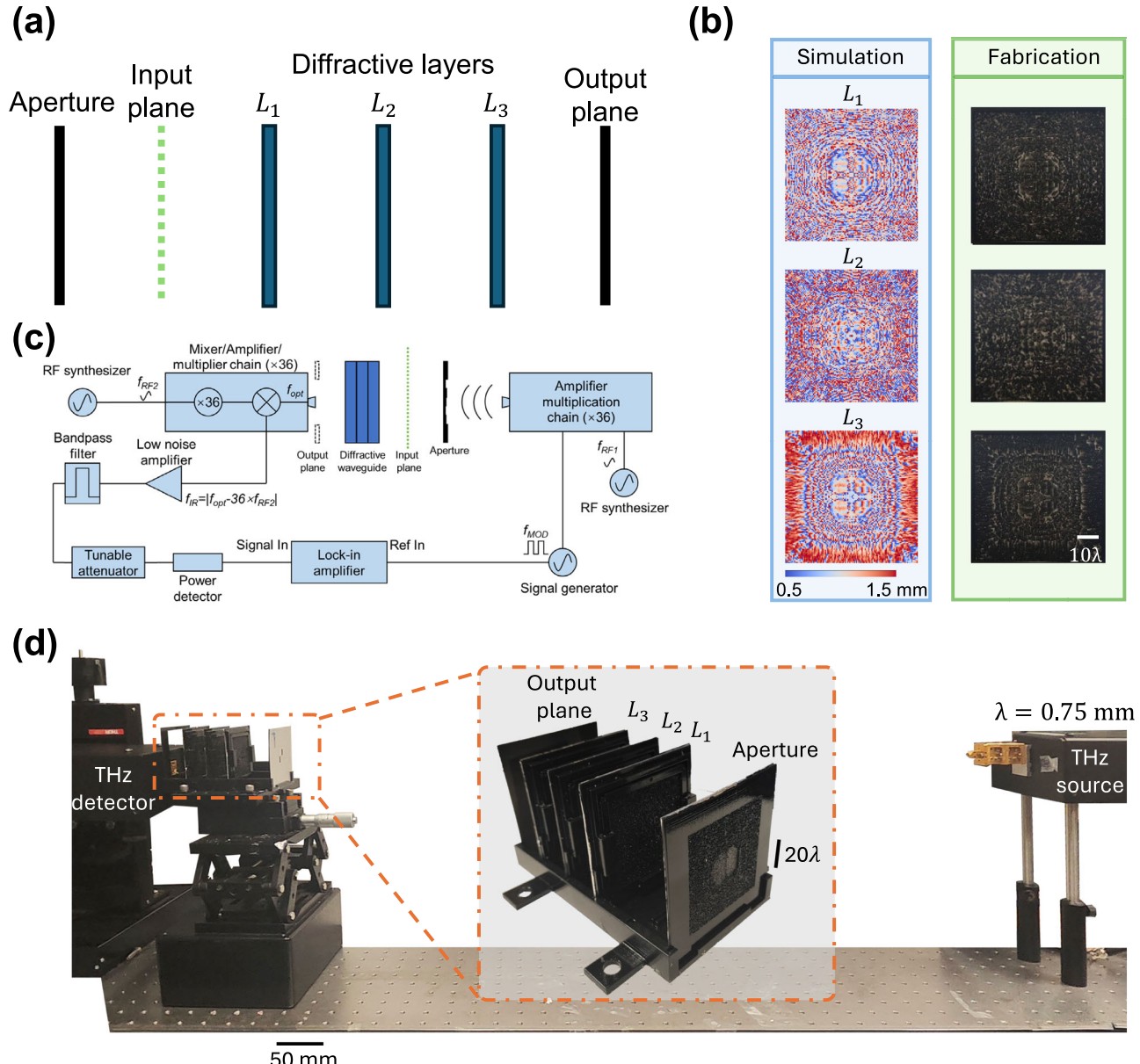

**Fig. 8 | Experimental validation of the mode filtering diffractive waveguide using terahertz radiation. a** Design schematic of a three-layer bandpass mode filtering diffractive waveguide, which passed $\{M_6, M_7\}$ and rejected the other spatial modes. **b** Left: Height profiles of the trained diffractive layers. Right: The fabricated diffractive layers used in the experiment. **c** Diagram of the terahertz scanning system. **d** Photograph of the experimental setup and the 3D-printed diffractive mode filter.

challenging to achieve with conventional designs. Another important distinction between our approach and prior work lies in the cascadability[55–57] of diffractive designs at high numerical apertures, e.g., NA = 1. Most metasurface architectures have approximation errors in their forward models at such a high NA, which can introduce cascaded errors that will build up as the length of the diffractive waveguide system increases. As a result, they are typically restricted to implementing a single function or an isolated task without a deeper cascaded architecture. In contrast, our diffractive waveguide platform operates by processing all the propagating modes within free space, i.e., with an NA of 1 in air, enabling modular cascading of transmissive diffractive surfaces, each contributing to a unified, end-to-end functionality. This task-specific cascadability, supported by our deep learning-based optimization and design pipeline, allows us to realize a wide range of spatial, spectral, and polarization-specific operations within a single integrated framework—highlighting

another key aspect of our work's novelty and utility beyond existing diffractive and metasurface architectures.

When compared to traditional imaging systems or 4-f systems, one of the key advantages of diffractive waveguides lies in the ability to offer far greater versatility and functionality beyond simple mode matching and guiding. Imaging-based systems and 4-f processors in general lack the capacity for more complex optical transformations that demand spatially varying programmed point spread functions. Based on universal linear transformations[58,59], diffractive waveguide designs enable a range of advanced functionalities, such as mode filtering, wavelength-dependent spatial and polarization mode operations, and multi-modal light processing, as demonstrated in Figs. 5–7. Although a 4-f system can offer some spatial mode manipulation through a programmable spatial filter placed at the Fourier plane, its design flexibility and degrees of freedom are considerably limited compared to our diffractive waveguide designs. As illustrated in

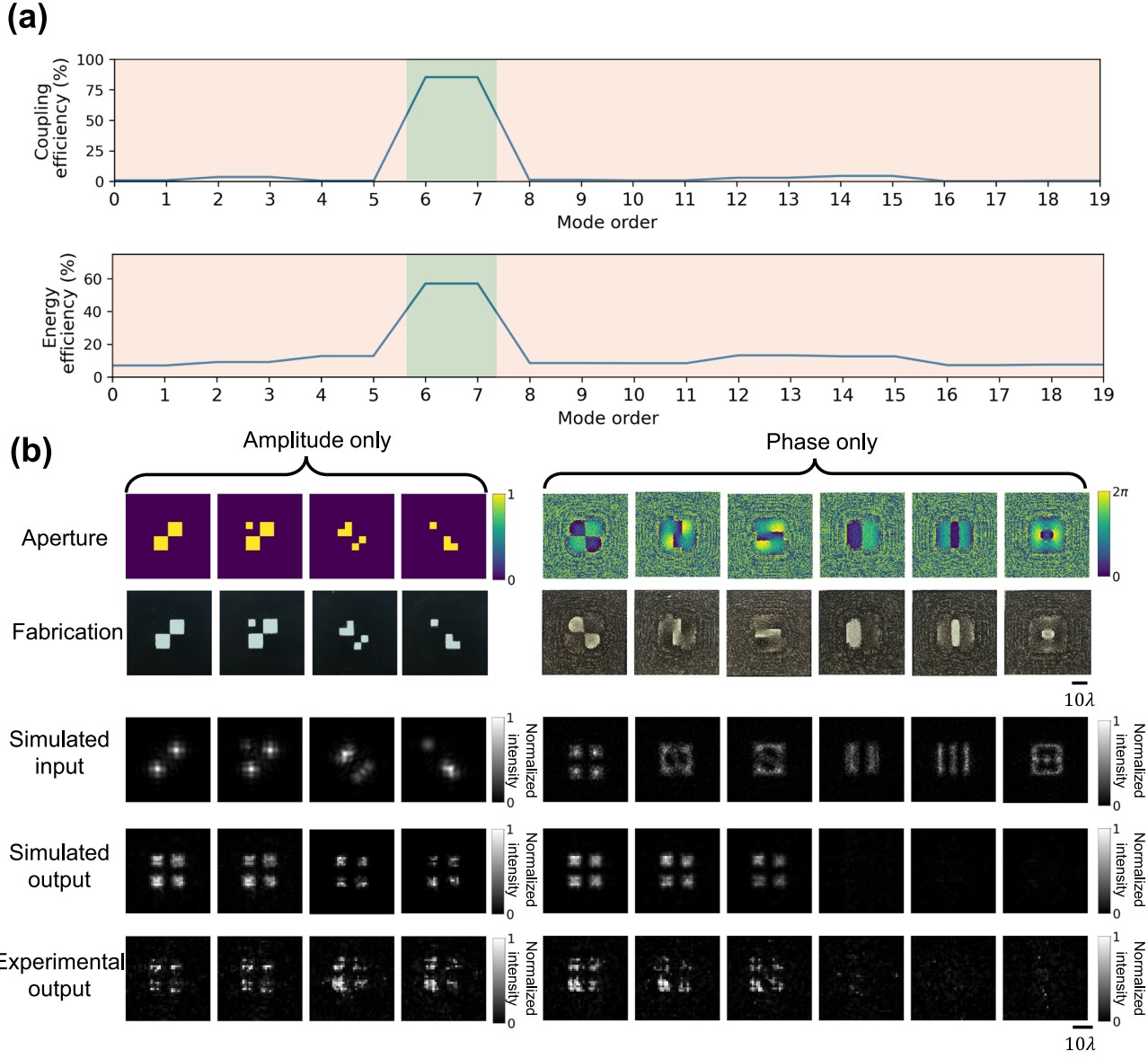

**Fig. 9 | Testing results of the bandpass mode filtering diffractive waveguide. a** Numerical analysis of the trained bandpass mode filtering diffractive waveguide in terms of coupling efficiency and energy efficiency. **b** Experimental results of the bandpass mode filtering diffractive waveguide, designed to pass only $\{M_6, M_7\}$.

Supplementary Fig. S15, we compared a programmable/trainable 4-f system optimized for mode filtering—using a trained complex-valued mask at the Fourier plane—with the performance of our mode filtering diffractive waveguides. These comparative results show that our diffractive waveguides outperform the programmable 4-f system for the same tasks, demonstrating superior functionality. Moreover, a 4-f system would require a significantly longer axial length (e.g., ~2400λ) to achieve the same NA as our design (which spans ~133λ). Therefore, diffractive waveguide designs allow for the selective transmission/rejection and processing of spatial, spectral, and polarization modes, on demand, by precisely controlling the optical characteristics of multiple diffractive layers within a compact volume. This versatility allows for the development of highly customized photonic systems for applications like mode-division and/or wavelength-division multiplexing, where the precise light manipulation beyond basic transmission is critical. Therefore, while imaging systems and programmable 4-f processors excel in achieving high transmission efficiency for simple optical mode profiles, they do not present the functional versatility or mode-specific advanced processing capabilities that our diffractive waveguide designs offer.

An important feature of our diffractive waveguide designs is their adaptability across different spectral bands. This versatility is particularly beneficial because the design framework can be scaled up or down to accommodate various spectral ranges by physically scaling the fabricated features proportional to the wavelength of interest. Consequently, transitioning between spectral bands does not necessitate re-training or redesigning the diffractive waveguide unit. This scalability not only simplifies the design process but also significantly reduces the time and resources needed to implement the designed diffractive waveguides for different applications, making them highly efficient to cover a wide range of optical technologies.

Compared to conventional spatial mode filtering systems, such as asymmetric Y-junctions[60], multi-mode interference couplers[61], and long-period gratings[62] our diffractive platform offers a number of distinct advantages. These conventional devices exhibit excellent performance in terms of coupling efficiency and mode rejection. However, extending their operation across a broad spectral band typically requires careful dispersion engineering, precise refractive index contrast, and complex fabrication processes. In contrast, our diffractive waveguide designs can be extended to operate

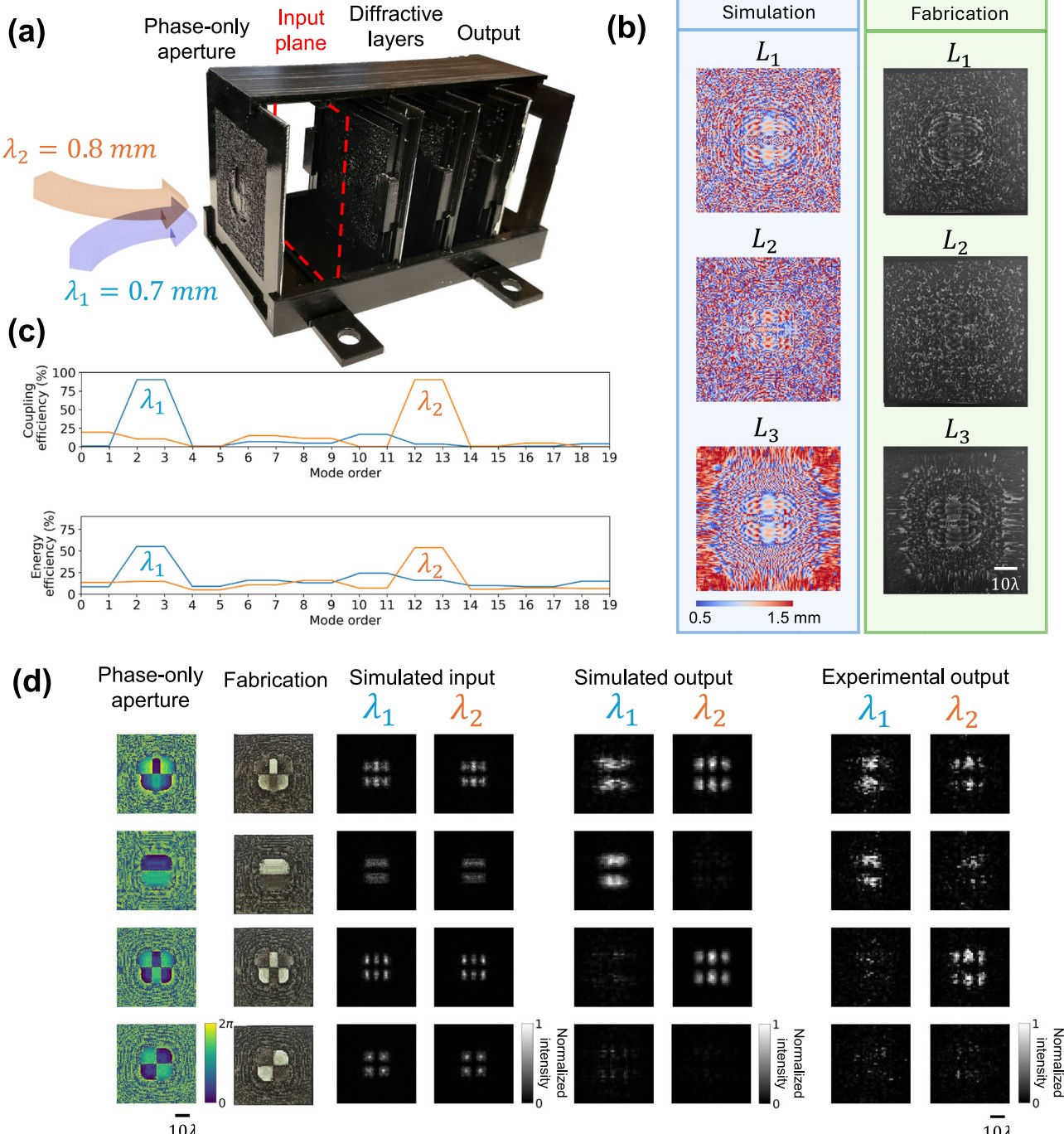

**Fig. 10 | Experimental validation of the multi-wavelength mode filtering diffractive waveguide. a** Photograph of the experimental setup and the 3D-printed multi-wavelength diffractive mode filter. **b** Left: Height profiles of the trained diffractive layers. Right: The fabricated diffractive layers used in the experiment. **c** Numerical analysis of the trained multi-wavelength mode filtering diffractive waveguide under different illumination wavelengths in terms of the coupling efficiency and the energy efficiency. **d** Experimental results of the multi-wavelength mode filtering diffractive waveguide, designed to pass $\{M_2, M_3\}$ at $\lambda_1 = 0.7$ mm and $\{M_{12}, M_{13}\}$ at $\lambda_2 = 0.8$ mm.

simultaneously at multiple wavelengths and polarization/mode combinations, performing a different desired function at each wavelength channel and polarization state or mode. For example, such a capability can be achieved by incorporating each wavelength channel with its associated desired task into the loss function during the deep learning-based training process[63]. Notably, the adaptation to multiple wavelength channels does not require material dispersion engineering. Instead, these layers can be constructed from low-loss materials, regardless of the dispersion properties of the diffractive materials[63], simplifying the design process and eliminating the dependency on the dispersion properties of the materials used. Therefore, the entire process of modal dispersion engineering within a diffractive waveguide can be handled by deep learning-based loss function optimization, as opposed to material engineering.

Operational versatility is another significant feature of our diffractive waveguides. They are capable of functioning effectively, whether situated in the air or immersed within various liquids or

gaseous environments. This adaptability opens up numerous applications in the field of sensing, where the ability to operate under different environmental conditions can enable accurate and reliable measurements of the properties of the medium that fills the space within the diffractive waveguide units. This flexibility greatly enhances the potential of our diffractive waveguides to be employed in a variety of real-world sensing applications, including the sensing of toxic gases and liquids, for example.

Another important feature of diffractive waveguides is their ability to preserve modes while filtering out undesired input noise. To assess the waveguiding performance of our diffractive designs under noisy input conditions, different levels of Gaussian noise were added to the input spatial modes, as shown in Supplementary Fig. S16. The Peak Signal to Noise Ratio (PSNR) metric was used to quantitatively evaluate the noise performance of diffractive waveguides (see the "Methods" section). The output results demonstrate that the diffractive waveguide presents a significant noise-filtering capability across different noisy conditions, increasing the PSNR of the output field compared to the noise-injected input modes. This filtering behavior further confirms the mode-preserving functionality of diffractive waveguides.

Although our analyses and results reported so far considered isotropic diffractive materials that are cascaded, diffractive waveguides can also be designed to maintain and multiplex different polarization states for different guided modes. To enable such a polarization-aware and polarization-multiplexed guiding of light through a diffractive waveguide, one can introduce a static 2D array of polarizers within the cascadable diffractive unit[38,45]. Such an architecture has been previously used to create universal polarization transformations to accurately perform thousands of spatially varying polarization scattering matrices at the diffraction limit of light[45]; this capability can be used as part of a cascadable diffractive waveguide unit, where appropriately engineered training loss functions associated with different polarization states can be assigned to different guided modes to convert the isotropic diffractive layers of a waveguide unit into a polarization-aware and polarization-multiplexed design.

The presented diffractive waveguides offer unique advantages over traditional waveguides, particularly in applications where selective mode, spectrum, and/or polarization manipulation of guided waves are critical. One prominent application is MDM in optical communication systems, where the ability to filter, process, and control spatial modes can significantly enhance data transmission capacity. Additionally, the wavelength-dependent programmable mode operations of our waveguides make them ideal for WDM systems, allowing for precise routing and filtering of optical signals based on their wavelength. Our compact designs also support integration into miniaturized photonic circuits, particularly beneficial in optical interconnects and other space-constrained systems. Furthermore, in optical signal processing, diffractive waveguides' mode, spectrum, and polarization filtering and manipulation capabilities can potentially improve microscopy, imaging, and holography systems. As another application area, in high-power laser systems, such diffractive waveguide designs can also enable advanced beam shaping and mode control, optimizing output beam quality. These diverse applications underscore the versatility and future potential of our diffractive waveguide framework in both established and emerging optics and photonics technologies.

Despite these advantages and important features, the fabrication process of diffractive waveguides presents several challenges that must be addressed. One such challenge is the limited phase bit depth of the diffractive layers, often constrained by the resolution of the 3D fabrication technique employed. Additionally, issues such as physical misalignments can degrade performance. To mitigate these challenges, we can incorporate the limitations of the fabrication and

alignment methods directly into the training process of the diffractive waveguide design. Specifically, the limited phase bit depth can be added as a design constraint during the training process such that the phase bit depth necessary for the effective operation of the diffractive waveguide can be reduced to e.g., 2–4 bits[36,39–41], as illustrated in Supplementary Fig. S6, which not only makes the fabrication process less demanding but also aligns well with the current manufacturing capabilities[41,63]. Furthermore, a "vaccination" strategy can be adopted to enhance the resilience of diffractive waveguides to unknown and random misalignments; this can be achieved by randomly shifting the diffractive layers of the waveguide design around their ideal positions during the training[36,37,39,40,44,54]. Such a vaccination strategy would effectively reduce the independent degrees of freedom within the diffractive waveguide design, which can be mitigated using wider or deeper diffractive unit architectures.

Another potential challenge for diffractive waveguides is multiple reflections that occur between adjacent diffractive layers. They were not a primary concern in this study as their effects are considerably weaker, and they are naturally filtered out by the cascaded diffractive layers, which collectively act as a spatial filter for such undesired secondary reflections that are not part of the training forward model. For example, the power transmission coefficient of a double reflected light is less than 0.5% in our experimental design, which is negligible compared to the >50% transmission coefficient of each layer, even with a lossy and inexpensive plastic material of a THz diffractive layer. For visible light applications, commonly used glass types such as BK7, with a refractive index of $n = 1.52$ and an extinction coefficient of $k = 9.75 \times 10^{-9}$, would offer much higher transmittance and lower reflectance, further supporting our conclusions that such secondary reflections do not constitute a practical challenge for diffractive waveguides. Furthermore, if needed, such secondary reflections can also be taken into account during the training stage at the cost of complicating and relatively slowing down the forward model of the diffractive waveguide design at the training epochs.

In addition, the maximum power a waveguide can handle is crucial in high-energy applications such as laser generation and optical power transmission. For GHz to THz wavelength operations, the diffractive layers in our waveguides can be made of polystyrene or other polymers, which have a relatively low melting point. However, this limitation is less significant at GHz to THz frequencies because the available power levels in these bands are inherently low. On the other hand, when the operating wavelength shifts to the visible or infrared spectrum, materials such as $SiO_2$, $TiO_2$, sapphire, or hydrogels can be used for the fabrication of diffractive layers[57,64]. Such materials offer greater thermal stability and are more resilient to high-energy operations, making them more suitable for applications that demand higher power handling capabilities.

Finally, our diffractive waveguides consist of several passive, cascadable structured layers that can be designed and optimized for diverse tasks with minimal adjustments to their configurations. This standardization offers a streamlined approach to task-specific waveguide designs. Besides, these waveguides can also be integrated with existing standard dielectric waveguides, including e.g., fiber optic cables, forming a modular set of components that can enhance the flexibility of conventional guided wave systems. The performance and design flexibility of these diffractive waveguides will stimulate further research and development, providing powerful solutions for a variety of applications, including telecommunications, imaging, sensing, and spectroscopy, among others.

## Methods
### Preparation of the spatial modes training datasets
In this work, we considered the guided spatial modes supported by a traditional square dielectric waveguide, which had an overall cladding

size of $80 \times 80$ mm and a central core area of $16 \times 16$ mm, where the wavelength of operation was selected as 0.75 mm. As shown in Fig. 1b, the refractive indices of the core and the cladding were set as $n_{core} = 1.6$ and $n_{cladding} = 1.4$, respectively. Based on this configuration, we calculated the spatial modes $\{M_0 \sim M_{39}\}$ supported by the traditional square dielectric waveguide. These spatial modes were calculated using a full-vectorial finite-difference method-based mode solver[46,47]. Due to the square cross-section, the computed spatial modes were symmetrical along the $x$ and $y$ directions. Additionally, given the isotropic nature of the waveguide, the modulation in different polarization directions remained the same. Consequently, we chose only the $x$-polarized optical wave for the model training and testing since the orthogonal polarization behaves the same way. With these calculated spatial modes $\{M_0 \sim M_{39}\}$ of the square dielectric waveguide, the input optical fields used in training were further enriched by linearly superposing various spatial modes, each with a random weight $w_i$ and a random bias phase $\varphi_i$ term ranging from 0 to $2\pi$. The resulting complex fields, denoted as $\psi_{\sup,Q}$, for $Q$ independent spatial modes, can be represented as:

$$\psi_{\sup,Q} = \sum_{i=1}^{Q} w_i \left( M_i \exp(j\varphi_i) \right) \tag{1}$$

where $M_*$ is the mode, and $j = \sqrt{-1}$.

## Numerical forward model of diffractive waveguides

In the design of a diffractive waveguide with $K$ diffractive layers, the forward model can be depicted by two successive operations: (i) free-space propagation of the optical field between two consecutive planes, and (ii) wave modulation by each diffractive layer. The free-space propagation for an axial distance $d$ of the complex field $u^l(x,y)$, immediately following the $l$th layer, was calculated using the angular spectrum approach[65], and can be written as:

$$\mathbb{P}_d u^l(x,y) = \mathcal{F}^{-1}\left\{ \mathcal{F}\left\{ u^l(x,y) \right\} H\left( f_x, f_y; d \right) \right\} \tag{2}$$

where $\mathbb{P}_d$ represents the free-space propagation operator for a distance $d$, $\mathcal{F}$ and $\mathcal{F}^{-1}$ are the two-dimensional Fourier transform and the inverse Fourier transform operators, respectively. $H\left( f_x, f_y; d \right)$ is the transfer function of free space, defined as:

$$H\left( f_x, f_y; d \right) = \begin{cases} \exp\left\{ jkd\sqrt{1 - \left( \frac{2\pi f_x}{k} \right)^2 - \left( \frac{2\pi f_y}{k} \right)^2} \right\}, & f_x^2 + f_y^2 < \frac{1}{\lambda^2} \\ 0, & f_x^2 + f_y^2 \geq \frac{1}{\lambda^2} \end{cases} \tag{3}$$

where $k = \frac{2\pi}{\lambda}$ and $\lambda$ is the wavelength of the light. $f_x$ and $f_y$ are the spatial frequencies along the $x$ and $y$ directions, respectively. The diffractive layers are modeled as thin optical elements that impart phase modulation to the incident field. The transmittance coefficient $t^l$ of the $l$th diffractive layer can be written as:

$$t^l(x,y) = \exp\left\{ j\phi^l(x,y) \right\} \tag{4}$$

where $\phi^l(x,y)$ is the phase modulation value of the trainable diffractive feature located at position $(x,y)$ of the $l^{\text{th}}$ diffractive layer. The final optical field at the output plane can be formulated as:

$$o(x,y) = \mathbb{P}_{d_{K,K+1}} \left( \prod_{l=1}^{K} t^l(x,y) \cdot \mathbb{P}_{d_{l-1,l}} \right) g(x,y) \tag{5}$$

where $d_{l-1,l}$ represents the axial distance between the $(l-1)^{\text{th}}$ and the $l^{\text{th}}$ diffractive layers, $g(x,y)$ is the input optical field.

In the simulation of the bent diffractive waveguide depicted in Fig. 4a, the tilted free-space propagation of the complex optical field was modeled using the angular spectrum approach and a coordinate rotation in the Fourier domains[51]. The optical field $u^l(x,y)$ following the $l^{\text{th}}$ layer, after propagating a distance of $d$, underwent a clockwise rotation by an angle $\theta$ to align with the new observation coordinate system. This process can be written as:

$$\mathbb{P}_{d,\theta} u^l(x,y) = \mathcal{F}^{-1}\left\{ G\left( \hat{f}_x(\theta), \hat{f}_y(\theta); d \right) \right\} \tag{6}$$

where $\mathbb{P}_{d,\theta}$ represents the operator of free-space propagation followed by a clockwise rotation. $G\left( \hat{f}_x(\theta), \hat{f}_y(\theta); d \right)$ is the Fourier domain representation of the optical field at the rotated observation plane, which is transformed from its counterpart $G\left( f_x, f_y; d \right)$ at the intermediate plane. This intermediate plane was parallel to the previous plane and located at a propagation distance $d$ away, which can be written as:

$$G\left( f_x, f_y; d, \phi \right) = \mathcal{F}\left\{ u^l(x,y) \right\} H\left( f_x, f_y; d \right) \tag{7}$$

The transformed spatial frequencies, $\hat{f}_x, \hat{f}_y$, are determined by the rotation angle $\theta$ of the observation plane and are linked by the transformation matrix in the Fourier domain.

$$k(x,y,z) = 2\pi \left( f_x, f_y, \left( \frac{1}{\lambda^2} - f_x^2 - f_y^2 \right)^{\frac{1}{2}} \right) \tag{8}$$

$$\hat{k}(x,y,z) = 2\pi \left( \hat{f}_x, \hat{f}_y, \left( \frac{1}{\lambda^2} - \hat{f}_x^2 - \hat{f}_y^2 \right)^{\frac{1}{2}} \right) \tag{9}$$

$$k = T^{-1}\hat{k} \tag{10}$$

where $k$ and $\hat{k}$ are the wave vectors of the intermediate plane and observation plane, respectively. The transformation matrix $T^{-1}$, which transforms the observation coordinates into the intermediate coordinates, can be formulated as:

$$T^{-1} = \begin{bmatrix} \cos\theta & 0 & \sin\theta \\ 0 & 1 & 0 \\ -\sin\theta & 0 & \cos\theta \end{bmatrix} \tag{11}$$

## Training loss functions and evaluation metrics

The diffractive waveguide units in this work were optimized using SGD-based supervised learning methods, which minimize a desired loss function tailored to the operational characteristics of the waveguide design. During the training of the diffractive waveguides shown in Fig. 1c, we used a single unit but also took into account its cascadability, as illustrated in Fig. 3a. To enhance the cascading capability of diffractive waveguides, a combined loss function was employed. This loss function integrates calculations of the complex input field $g$ and the output complex field $o_i$ of several identical waveguide units, where $i = 1, 2, 3 \ldots, N$. The corresponding loss function was defined as:

$$\text{loss}(g, o_i) = \alpha_1 (1 - E(g, o_1)) + \sum_{i=1}^{N} \beta_i (1 - \text{CE}(g, o_i)) \tag{12}$$

where $E(\cdot)$ and $\text{CE}(\cdot)$ denote the energy efficiency and the coupling efficiency, respectively. $\alpha_1$ and $\beta_1, \beta_2, \ldots, \beta_N$ are empirical constants that adjust the relative weights of the different loss terms. The energy efficiency function measures the energy ratio of the output field $o$ to the input complex field $g$, which is defined as:

$$E(g, o) = \frac{\sum oo^*}{\sum gg^*} \tag{13}$$

The coupling efficiency was calculated using[66]:

$$\text{CE}(g, o_i) = \frac{\left| \iint g o_i^* dx dy \right|^2}{\iint g g^* dx dy \iint o_i o_i^* dx dy} \quad (14)$$

To simplify the training process, the coupling efficiency loss was only considered for $i = \{1, 2, 3\}$ and the hyperparameters $(\alpha_1, \beta_1, \beta_2, \beta_3)$ were empirically set to $(1, 10, 10, 5)$.

A similar loss function was used for training the bent diffractive waveguide in Fig. 4a, while only the output after the first unit was considered. This function can be expressed as:

$$\text{loss}(g, o) = \alpha(1 - \text{CE}(g, o)) + \beta(1 - E(g, o)) \quad (15)$$

The constants $(\alpha, \beta)$ were empirically set as $(2, 1)$.

For the designs shown in Supplementary Fig. S10a, the mode filtering diffractive waveguide was trained using a dual-component loss function aimed at maximizing the transmission of target modes $(\text{loss}^+)$ and minimizing the transmission of rejected modes $(\text{loss}^-)$, defined as:

$$\text{loss} = \text{loss}^+ + \text{loss}^- \quad (16)$$

where $\text{loss}^+$ was defined by:

$$\text{loss}^+ = \alpha_1 \text{ReLU}\left(T_{\text{CE},u} - \text{CE}(g^+, o^+)\right) + \beta_1 \text{ReLU}\left(T_{E,u} - E(g^+, o^+)\right) \quad (17)$$

and $\text{loss}^-$ was given by:

$$\text{loss}^- = \alpha_2 \text{ReLU}\left(\text{CE}(g^-, o^-) - T_{\text{CE},l}\right) + \beta_2 \text{ReLU}\left(E(g^-, o^-) - T_{E,l}\right) \quad (18)$$

where $(\cdot)^+$ and $(\cdot)^-$ indicate the target and rejected modes, respectively. ReLU$(\cdot)$ is the rectified linear unit function. $T_{\text{CE},u}$, $T_{E,u}$, $T_{\text{CE},l}$, $T_{E,l}$ are the upper and lower thresholds for coupling efficiency and energy efficiency, respectively, which were set as $(0.9, 0.5, 0.05, 0.05)$. The hyperparameters $\alpha_1$, $\alpha_2$, $\beta_1$, $\beta_2$ were empirically set as $(1, 1, 1, 1)$. For the designs shown in Fig. 5a, the multi-wavelength mode filtering diffractive waveguide design was optimized using the same loss function as the monochrome mode filtering diffractive waveguide, with the same hyperparameter set.

In the mode splitting diffractive waveguide design, we used $Q = 4$ modes in the superimposed input fields $\psi_{\text{sup},4}$, and the output fields were decomposed into the corresponding modes by calculating the coupling efficiency of the output ($o$) and each target spatial mode ($M_{i=1,..,Q}$). The diffractive layers were trained to minimize the loss function:

$$loss = \alpha \sum_{i=1}^{4} \text{MSE}(\text{CE}(o, M_i), w_i) + \beta \left( E\left(o, \sum_{i=1}^{4} M_i\right) \right) \quad (19)$$

where $w_i$ was the presumed weight for the modal superposition shown in Eq. 1, MSE$(\cdot)$ is the MSE. $(\alpha, \beta)$ were empirical set as $(10, 1)$. The loss function for multi-wavelength mode splitting waveguide design was similar to the monochrome version, with an additional loss term that equalized the energy performance among different wavelength channels:

$$loss = \alpha \sum_{i=1}^{4} \text{MSE}(\text{CE}(o, M_i), w_i) + \beta \left( E\left(o, \sum_{i=1}^{4} M_i\right) \right)$$
$$+ \gamma \text{abs}\left( \sum o_{\lambda_1} o_{\lambda_1}^* - \sum o_{\lambda_2} o_{\lambda_2}^* \right) \quad (20)$$

where abs$(\cdot)$ is the absolute value, $o_{\lambda_*}$ is the optical field illuminated at a wavelength $\lambda_*$. $(\alpha, \beta, \gamma)$ were empirically set as $(10, 1, 1)$.

In the mode-specific polarization-maintaining diffractive waveguide design, the diffractive model was trained by maximizing the target mode transmission at the correct polarization direction while minimizing other mode transmission, defined as:

$$\text{loss} = \alpha\left(1 - \text{CE}(o^+, g^+) + \text{CE}(o^-, g^-)\right) + \beta\left(1 - E(o^+, g^+) + E(o^-, g^-)\right) \quad (21)$$

where $(\cdot)^+$ and $(\cdot)^-$ indicate the target and rejected modes, respectively. The constants $(\alpha, \beta)$ were empirically set as $(1, 4)$.

In assessing the denoising capability of diffractive waveguides, the PSNR metric was calculated by:

$$\text{PSNR} = 10 \log_{10}\left( \frac{\text{MAX}^2}{\text{MSE}(I, J)} \right) \quad (22)$$

where MAX is the maximum intensity of the output. $I$ is the noise-free output, and $J$ is the noisy output.

The cross-correlation metric used to evaluate the similarities between our numerical simulations and experimental results was calculated using:

$$r(s, e) = \frac{\sum_m \sum_n (s_{mn} - \bar{s})(e_{mn} - \bar{e})}{\sqrt{\left( \sum_m \sum_n (s_{mn} - \bar{s})^2 \right)\left( \sum_m \sum_n (e_{mn} - \bar{e})^2 \right)}} \quad (23)$$

where $s$ are the numerical simulation results and $e$ are the experimental results, $\bar{s}$ and $\bar{e}$ are the average values of $s$ and $e$, respectively.

### Parameters of digital implementation and training scheme

For the square diffractive waveguide, bent diffractive waveguide, and the mode filtering diffractive waveguide, each diffractive layer contained $200 \times 200$ diffractive features, each with a size of 0.4 mm $(0.53\lambda)$, dedicated solely to phase modulation. For the mode splitting diffractive waveguide design, the diffractive layers consisted of $500 \times 200$ (monochromatic) and $500 \times 500$ (multi-wavelength) diffractive features with a pixel size of 0.4 mm $(0.53\lambda)$. In the square diffractive waveguide, the axial distance between the input plane or output plane and the nearest diffractive layer, as well as between any two adjacent diffractive layers, was set to be 13.33 mm $(17.78\lambda)$. For the bent diffractive waveguide, the center-to-center distance between two successive planes, including the input plane, output plane, and diffractive layers, was set to be 20 mm $(26.67\lambda)$. The first diffractive layer was parallel to the input plane, and the output plane was parallel to the last diffractive plane. Starting from the second diffractive layer, each layer was rotated 15° clockwise relative to the normal direction of its predecessor, adjusting the direction of light propagation. For the mode filtering diffractive waveguide, similar settings were used, with an axial distance of 20 mm $(26.67\lambda)$ for the monochrome version and 14.28 mm $(19.04\lambda)$ for the multi-wavelength version between either the input or output planes and the nearest diffractive layer, as well as between any two successive diffractive layers. In the design of the mode splitting diffractive waveguide, a 40 mm $(53.33\lambda)$ gap was used between the output FOV of the two output channels. The axial distance was also 20 mm $(26.67\lambda)$ between any two adjacent planes. As for the multi-wavelength mode splitting diffractive waveguide design, a 40 mm $(53.33\lambda)$ crossing gap was used at the center of the output FOV to separate four output channels. The axial distance between any two adjacent layers was set to be 5 mm $(6.67\lambda)$. For the mode-specific polarization-maintaining diffractive waveguide design, the axial distance between any two successive layers was set to be 2 mm $(2.67\lambda)$. The PA consists of $200 \times 200$ linear polarizers of four different polarization directions. These 4 different types of linear polarizers are spatially binned to have a $2 \times 2$ period and repeated with 50 periods in each direction. The period of each linear PA is $2.14\lambda$.

The numerical simulations and the training process for the diffractive waveguides described in this study were carried out using Python (version 3.11.15) and PyTorch (version 2.1.2, Meta Platform Inc.). The Adam optimizer[67] with its default settings was utilized. We set the learning rate at 0.0002 and the batch size at 128. Our diffractive models underwent a 600-epoch training on a workstation equipped with an Nvidia GeForce RTX 4090 GPU, an Intel Core i9−13900KF CPU, and 32 GB RAM. The training time for a diffractive waveguide design was roughly 3 h, which is a one-time design effort.

### Experimental design and testing

In our experimental setup, we explored the implementation of a mode filtering diffractive waveguide design by fabricating and assembling the diffractive layers using a 3D printer (Objet30 Pro, Stratasys) and testing it with a CW source at $\lambda = 0.75$ mm (shown in Fig. 8a, d). For these experiments, we trained a three-layer mode filtering diffractive waveguide using the same configuration as the system reported in Supplementary Fig. S10, with the following changes: (i) the input, output FOV and diffractive layers of the diffractive mode filter were reduced to $48 \times 48$ mm ($64\lambda \times 64\lambda$), with the same virtual core region of $16 \times 16$ mm; (ii) the hyperparameters for the loss function were adjusted to $(\alpha_1, \alpha_2, \beta_1, \beta_2) = (1, 1, 6, 1)$, $(T_{CE,u}, T_{E,u}, T_{CE,l}, T_{E,l}) = (0.85, 0.5, 0.05, 0.05)$; and (iii) the target mode set was defined to be $S_t = \{M_6, M_7\}$ and the reject set contains the rest of the guided modes. The input apertures used in the experiments included either amplitude-only or phase-only modulation of the input light field to generate various complex optical fields at the input plane. For amplitude-only input apertures, we uniformly segmented the central part of the aperture into 16 patches, each measuring $4 \times 4$ mm. These amplitude-only input apertures were selected from all the binary combinations of either passing or blocking light in these patches to maximize the ratio of the target modes $S_t$ at the input plane. The phase-only input apertures were optimized using SGD to maximize or minimize the ratio of the target modes $S_t$ at the input plane.

As shown in Fig. 8c, the experimental evaluation of the 3D-printed diffractive waveguide employed a terahertz scanning system. A modular amplifier (Virginia Diode Inc. WR9.0 M SGX) and multiplier chain (Virginia Diode Inc. WR4.3 × 2 WR2.2 × 2) (AMC) with a compatible diagonal horn antenna (Virginia Diode Inc. WR2.2) were employed as the THz source. A 10 dBm RF input signal at 11.1111 GHz ($f_{RF1}$) was fed into the input AMC, which was then multiplied 36 times to generate the CW radiation at 0.4 THz ($f_{opt}$). Additionally, the AMC was modulated with a 1 kHz square wave for lock-in detection. The input plane of the 3D-printed diffractive waveguide was positioned about 75 cm away from the horn antenna's exit aperture, ensuring an approximately uniform plane wave incident on its input FOV. After passing through the diffractive waveguide, the output signal was 2D scanned with an 8 mm step using a single-pixel mixer (Virginia Diode Inc. WRI 2.2) mounted on an XY positioning stage assembled from two linear motorized stages (Thorlabs NRT100). A 10 dBm RF signal at 11.0833 GHz ($f_{RF2}$) was sent to the detector as a local oscillator to down-convert the received signal to 1 GHz ($f_{IR}$). The down-converted signal was then amplified by a low-noise amplifier (Mini-Circuits ZRL-1150-LN+) and filtered by a 1 GHz (±10 MHz) bandpass filter (KL Electronics 3C40−1000/T10-O/O). After the filtration, the signal was passed through a low-noise power detector (Mini-Circuits ZX47-60) and then measured by a lock-in amplifier (Stanford Research SR830) with a 1 kHz square wave serving as the reference signal. The readings from the lock-in amplifier were calibrated to a linear scale.

### Data availability
All the data and methods needed to evaluate the conclusions of this work are presented in the main text and Supporting Information.

Additional data can be requested from the corresponding author (A.O.).

### Code availability
The codes used in this work use standard libraries and scripts that are publicly available in PyTorch.

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

## Acknowledgements

Ozcan Research Lab at UCLA acknowledges the funding of the U.S. ARO (Army Research Office). Jarrahi Lab acknowledges the support of U.S. DOE (Department of Energy, DE-SC0016925).

## Author contributions

A.O. conceived and initiated the research. Y.W., Y.L., and K.L. discussed the initial implementations. Y.W., Y.L., and T.G. conducted the experiments and processed the resulting data. Y.W. and Y.L. conducted numerical simulations. All the authors contributed to the preparation of the manuscript. A.O. and M.J. supervised the research.

## Competing interests

A.O., Y.W., and Y.L. are co-inventors of a pending patent application on the presented method. M.J. is a co-founder of Lookin Inc. The remaining authors declare no competing interests.
