## [Transparent Peer Review file · Nature Communications]

Optimizing Structured Surfaces for Diffractive Waveguides

Corresponding Author: Dr Aydogan Ozcan

Version 0:

Reviewer comments:

Reviewer #1

(Remarks to the Author)

The authors present the design and optimization of diffractive waveguides using deep learning, validated through numerical simulations and experimental demonstrations in the terahertz (THz) regime. As a fan of research that combines intelligent hardware and advanced machine learning, I find this paper to be an interesting and novel contribution, showcasing a novel application of multiple diffractive layers to form waveguides.

The study demonstrates various functionalities, including mode filtering, splitting, and bent waveguides, by exploring diffractive waveguides with diverse structures. In addition, the scalability and modularity of the designs indicate the potential for a wide range of applications. Importantly, experimental validation at THz frequencies supports the feasibility of the proposed designs, achieving selective mode transmission and rejection through 3D-printed diffractive layers.

The paper is well-written, logically organized, and supported by multiple figures that effectively illustrate the modeling results and experimental findings. This work offers a compelling alternative to conventional waveguides, and I highly recommend it for publication in Nature Communications.

Minor comments:

1. It may be beneficial to include additional details of the work in the title, emphasizing key aspects such as the design of diffractive waveguides, the role of deep learning in optimization, and the experimental validation conducted in the THz regime. I leave this to the authors' discretion.
2. Providing an explanation of how the phase modulation patterns in Figure 1d facilitate mode matching would improve the understanding of the functionality of the diffractive waveguide.
3. In Figure 1e, it would be helpful to include quantitative values (e.g. relative error or coupling efficiency metrics) to specify the degree of difference between the output fields of the traditional dielectric waveguide and the proposed diffractive waveguide. This addition would clarify and strengthen the claim that the diffractive waveguide closely matches the performance of its traditional counterpart.
4. The efficiency drops in cascaded structures and higher-order modes are briefly mentioned for both normal and bent diffractive waveguides. A more detailed analysis of the contributing factors such as mode distortions, waveguide material properties, or potential mitigation strategies would be helpful to support the practical feasibility of the proposed design.
5. The experimental validation of the diffractive waveguide at THz frequencies, utilizing practical 3D printing and measurement techniques, demonstrates its feasibility. Discussing how these results could be extended to other spectral ranges, such as visible or infrared, would be valuable. These ranges may present unique challenges, including fabrication complexities, environmental variations, and noise, which merit further exploration.
6. The testing results of the bent diffractive waveguides at $K = 5$ are different for coupling and energy efficiency between Supplementary Figures S4 and S5. It would be necessary to clarify this.

Reviewer #2

(Remarks to the Author)

The authors use deep learning to develop diffractive patterns that can be used as an alternative to traditional refractive waveguides that rely on refractive index contrast. This framework is also capable of producing waveguides with specialized functionalities such as propagation rotation, mode splitting and filtering, and polarization maintenance. The authors compare their diffractive waveguide with a traditional square refractive waveguide showing low propagation loss and high coupling efficiency. Their work benefits from the lack of material dispersion in diffractive optics. In my judgement, this demonstration and methodology for developing diffractive optics are significant additions to the literature and will have interest among a wide readership. Overall, the work provides extensive demonstrations of the benefits of the diffractive waveguide system and makes a compelling case for its use over conventional systems. Nonetheless, it is my assessment that the work would benefit from a few additional experiments and evidence as outlined below. Further, I believe the manuscript should also clarify a few points listed below to improve the scientific quality of the material.

1. A key advantage of refractive processes over diffractive ones is higher efficiency. Can the authors provide a quantification of the energy loss from their diffractive system compared to the equivalent refractive dielectric square waveguide? While the amplitude and phase profiles present only small relative errors, energy loss is expected to be higher in the authors' diffractive system, so a quantification of that would be a valuable addition.
2. In Figure 2, the authors provide a visualization of the input and output fields generated from their model. There is a large drop in the coupling efficiency for modes M30-37. This drop casts doubt on the model's ability for external generalization as the decrease occurs primarily for the modes that were not used in the training set. The authors should explain this drop in coupling and provide evidence of whether this is a physical result, or an artifact of their model's limitations.
3. The authors claim that their system could be adapted for on-chip optical circuits. However, I am concerned about the feasibility of such applications given the massive reduction in feature size required for on-chip applications. Can the authors provide some estimated feature size required for an on-chip version of their design to support these statements?
4. The demonstrating of spatial mode filtering is quite convincing and impressive; however, in my opinion, it would benefit from benchmarking to conventional spatial mode filters in terms of rejection ratios and coupling efficiencies.
5. My main concern with this work is its novelty compared to the extensive literature on diffractive optics and meta lenses. Many of the functionalities demonstrated here, mode splitting, and spatial mode filtering have been demonstrated in prior works (Khonina, Svetlana N., et al. "Advancements and applications of diffractive optical elements in contemporary optics: A comprehensive overview." *Advanced Materials Technologies*(2024): 2401028.). Thus, the main novelty seems to be the deep learning algorithm developed by the authors and the modularity of their system. My suggestion is that the authors clarify the novelty and uniqueness of their work by comparing it to similar prior works. Any experiments demonstrating this novelty or an improvement in optical parameters would bolster the case for diffractive waveguides.

Following the above recommended changes, I judge that the manuscript will be ready for publication.

Version 1:

Reviewer comments:

Reviewer #1

(Remarks to the Author)

The authors have revised the paper in accordance with the reviewers' suggestions and comments.

Reviewer #2

(Remarks to the Author)

I am satisfied with the authors' responses to my comments and recommendations. I believe this work can be published in its current form.

We sincerely thank the referees for their reviews and the constructive feedback that we have received on our manuscript “**Optimizing Structured Surfaces for Diffractive Waveguides**” submitted to *Nature Communications* (Manuscript ID: NCOMMS-24-80435).

As detailed below, we have revised our manuscript in response to the reviewers’ comments. The original referee comments are shown in black color, whereas for ease of communication, our answers are provided in blue. Our revisions have also been marked in the main text and supplementary information files using yellow highlighting.

Summary of our Revisions:

In addition to improving our manuscript based on the referees’ feedback, we have added new results and discussion in the **revised manuscript and Supplementary Information files**. As a quick summary, the following items have been revised and added to our manuscript's main text and SI, highlighted in yellow:

Revised sub-sections:

- **Changes to text:**
 - Title
 - Results
 - Discussion
 - Supplementary Information

Supplementary Figures Revised:

- **Supplementary Fig. S9**

New Supplementary Figures Added:

- **Supplementary Fig. S1. Cross sectional profile of mode propagation within the diffractive waveguide shown in Fig. 1d of main text.**
- **Supplementary Fig. S2. Testing results of diffractive waveguide models trained with different sets of spatial modes.** Five sets were used for training: I. 20 lower order modes ($M_0 \sim M_{19}$, same as Fig. 2); II. all 40 modes ($M_0 \sim M_{39}$); III. 20 higher order modes ($M_{20} \sim M_{39}$); IV. a combination of 10 lower order modes $M_0 \sim M_9$ and 10 higher order modes $M_{30} \sim M_{39}$; V. two consecutive modes followed by skipping two, i.e., $\{M_0, M_1, M_4, M_5, M_8, M_9, \dots, M_{32}, M_{33}, M_{36}, M_{37}\}$. All the trained diffractive waveguide models were tested with the full set of 40 spatial modes.
- **Supplementary Fig. S3. Testing results of the diffractive waveguide models trained with and without incorporating material absorption into the training process.** Both diffractive waveguide models were tested with an absorbing material; we assumed $\lambda = 0.75$ mm and the complex refractive index of the diffractive layers was assumed to be $1.72 + j 0.03$, which corresponds to a commonly used 3D printing material (VeroBlack, Objet30 Pro, Stratasys). Lower loss materials can be used to further improve the energy efficiency of these diffractive waveguide designs.
- **Supplementary Fig. S6. Testing results of a single-mode diffractive waveguide operating at 1550 nm, designed with varying levels of phase quantization bit depths.** (a) Input field,

output field and diffractive layer phase profiles corresponding to different phase quantization bit depths. (b) Coupling and energy efficiency values of diffractive waveguides designed with different phase quantization bit depths.

Reviewer 1:

The authors present the design and optimization of diffractive waveguides using deep learning, validated through numerical simulations and experimental demonstrations in the terahertz (THz) regime. As a fan of research that combines intelligent hardware and advanced machine learning, I find this paper to be an interesting and novel contribution, showcasing a novel application of multiple diffractive layers to form waveguides.

The study demonstrates various functionalities, including mode filtering, splitting, and bent waveguides, by exploring diffractive waveguides with diverse structures. In addition, the scalability and modularity of the designs indicate the potential for a wide range of applications. Importantly, experimental validation at THz frequencies supports the feasibility of the proposed designs, achieving selective mode transmission and rejection through 3D-printed diffractive layers.

The paper is well-written, logically organized, and supported by multiple figures that effectively illustrate the modeling results and experimental findings. This work offers a compelling alternative to conventional waveguides, and I highly recommend it for publication in Nature Communications.

-- We sincerely thank the referee for their valuable assessment and constructive feedback, which helped us further enhance the quality and clarity of our manuscript.

Minor comments:

1. It may be beneficial to include additional details of the work in the title, emphasizing key aspects such as the design of diffractive waveguides, the role of deep learning in optimization, and the experimental validation conducted in the THz regime. I leave this to the authors' discretion.

-- Following the referee's suggestion, we have revised the title to:
Optimizing Structured Surfaces for Diffractive Waveguides.

2. Providing an explanation of how the phase modulation patterns in Figure 1d facilitate mode matching would improve the understanding of the functionality of the diffractive waveguide.

-- We thank the reviewer for raising this discussion. Accordingly, we have added a new figure, **Supplementary Fig. S1**, to show the cross-sectional light field distribution within the diffractive waveguide illustrated in **Fig. 1d**. The newly added figure demonstrates that optimized diffractive layers exhibit phase modulation patterns structured at the diffraction limit of light, that sequentially relay the input light as it passes through the successive layers. Furthermore, **Supplementary Fig. S7c** shows the light field evolution within the diffractive waveguide over a longer propagation distance, further supporting the same picture.

A new figure, **Supplementary Fig.S1**, was added to the revised Supplementary Information file and was discussed in the revised main text, as quoted below:

“... The optimized diffractive layers exhibit phase modulation patterns, structured at the diffraction limit of light, that sequentially relay the input light as it passes through the successive layers. The corresponding cross-sectional light field profile is also shown in **Supplementary Fig. S1.**”

Supplementary Fig. S1. Cross sectional profile of mode propagation within the diffractive waveguide shown in Fig. 1d of main text.

3. In Figure 1e, it would be helpful to include quantitative values (e.g. relative error or coupling efficiency metrics) to specify the degree of difference between the output fields of the traditional dielectric

waveguide and the proposed diffractive waveguide. This addition would clarify and strengthen the claim that the diffractive waveguide closely matches the performance of its traditional counterpart.

-- In our study, we evaluated the performance of diffractive waveguides using both the **coupling efficiency** and **energy efficiency**, as reported in **Fig. 2b**. For traditional dielectric waveguides, intrinsic attenuation losses can arise from, e.g., scattering and absorption. However, in our analysis, the dielectric waveguides used for comparison were assumed to have ideal refractive index profiles without any material defects and structural inhomogeneities. This assumption effectively eliminates various sources of losses for the standard dielectric waveguides considered in our comparison. Under these conditions, coupling and energy efficiencies are set to 100% for the traditional dielectric waveguides, serving as a reference baseline for evaluating the performances of our diffractive waveguide designs. These points have been incorporated into the **Results** section of our revised manuscript, as quoted below:

“... In our analysis, the dielectric waveguide used for comparison was assumed to have an ideal refractive index profile, without any material defects and structural inhomogeneities. This assumption effectively eliminates various sources of losses for the standard dielectric waveguides considered in our comparison. Under these conditions, coupling and energy efficiencies are set to 100% for the traditional dielectric waveguides, serving as a reference baseline for evaluating the performances of our diffractive waveguide designs.”

In Figure 1(e), the output fields from the square dielectric waveguide and the diffractive waveguide are also compared, with the relative errors displayed (at the bottom), which are negligible. The output fields of the traditional dielectric waveguide were calculated by full-vectorial finite-difference method, serving as the ground truth for evaluating the output fields of the diffractive waveguide.

4. The efficiency drops in cascaded structures and higher-order modes are briefly mentioned for both normal and bent diffractive waveguides. A more detailed analysis of the contributing factors such as mode distortions, waveguide material properties, or potential mitigation strategies would be helpful to support the practical feasibility of the proposed design.

-- We thank the reviewer for raising the discussion regarding potential reasons and mitigation methods for the efficiency drop in cascaded structures and for higher-order modes. Accordingly, we have tested the coupling and energy efficiency of diffractive waveguides trained with different sets of spatial modes, as shown in the **newly added Supplementary Fig.S2**. In this new analysis, five diffractive waveguide models were trained with: I. 20 lower order modes ($M_0 \sim M_{19}$, same as Fig. 2); II. all 40 modes ($M_0 \sim M_{39}$); III. 20 higher order modes ($M_{20} \sim M_{39}$); IV. 10 a mixed set of lower order modes $M_0 \sim M_9$ and higher order modes $M_{30} \sim M_{39}$; V. an alternating pair based sampling: $\{M_0, M_1, M_4, M_5, M_8, M_9, \dots, M_{32}, M_{33}, M_{36}, M_{37}\}$. All models were tested with the full set of 40 spatial modes. **The second model, which was trained with the full set of modes, achieved the highest coupling efficiency values, while also retaining very high energy efficiency, as illustrated in Supplementary Fig.S2.**

These results confirm that the observed drop in coupling efficiency for $M_0 \sim M_{37}$ in Fig. 2 is not due to a limitation of the model architecture or the optimization process, but rather due to the absence of those modes in the training set, which can be significantly improved by including these modes in the training stage as demonstrated in the **new Supplementary Fig. S2**.

Furthermore, to account for material absorption in experimental scenarios, we have introduced an additional analysis, as shown in the **newly added Supplementary Fig. S3**, where the training process incorporated material losses. The comparison between diffractive waveguides trained with and without material absorption reveals that coupling efficiency and energy efficiency can be improved significantly by taking into account material-induced losses in the training process.

Accordingly, two new figures, i.e., **Supplementary Fig. S2** and **Fig. S3**, were added to the revised **Supplementary Information** and these new analyses were added to the **Results** section, quoted below:

*“...To further investigate the coupling efficiency drop observed for higher order modes, we analyzed the performance of diffractive waveguides trained on different spatial mode sets (see **Supplementary Fig.S2**). These results show that the coupling efficiency values are significantly improved when all the desired spatial modes of interest are known and accessible during training. Additionally, to account for material absorption in practical implementations, we incorporated energy absorption into the training process, as shown in **Supplementary Fig.S3**. Compared to designs trained without considering material absorption, those optimized with absorption exhibited improved efficiency when tested with absorbing diffractive materials, demonstrating the benefit of accounting for such factors in the training loss function and optimization process.”*

Supplementary Fig. S2. Testing results of diffractive waveguide models trained with different sets of spatial modes. Five sets were used for training: I. 20 lower order modes ($M_0 \sim M_{19}$, same as in Fig. 2); II. all 40 modes ($M_0 \sim M_{39}$); III. 20 higher order modes ($M_{20} \sim M_{39}$); IV. a combination of 10 lower order modes $M_0 \sim M_9$ and 10 higher order modes $M_{30} \sim M_{39}$; V. two consecutive modes followed by skipping two, i.e., $\{M_0, M_1, M_4, M_5, M_8, M_9, \dots, M_{32}, M_{33}, M_{36}, M_{37}\}$. All the trained diffractive waveguide models were tested with the full set of 40 spatial modes.

Supplementary Fig. S3. Testing results of the diffractive waveguide models trained with and without incorporating material absorption into the training process. Both diffractive waveguide models were tested with an absorbing material; we assumed $\lambda = 0.75$ mm and the complex refractive index of the diffractive layers was assumed to be $1.72 + j 0.03$, which corresponds to a commonly used 3D printing material (VeroBlack, Objet30 Pro, Stratasys). Lower loss materials can be used to further improve the energy efficiency of these diffractive waveguide designs.

5. The experimental validation of the diffractive waveguide at THz frequencies, utilizing practical 3D printing and measurement techniques, demonstrates its feasibility. Discussing how these results could be extended to other spectral ranges, such as visible or infrared, would be valuable. These ranges may present unique challenges, including fabrication complexities, environmental variations, and noise, which merit further exploration.

-- To address these important points of the referee, we have added a new figure, i.e., **Supplementary Fig. S6**, along with the following text to the **Discussion** part of the revised manuscript:

*“... To explore the feasibility of implementations in the near-infrared part of the spectrum, we conducted simulations at 1550 nm wavelength for approximating the waveguiding behavior of SMF-28 fiber⁴⁸, which is a single-mode optical fiber that is widely used in telecommunications. As shown in **Supplementary Fig. S6**, >95% coupling efficiency and >95% energy efficiency can be achieved with 4-bit depth diffractive layers ($K = 2$), with each layer having a lateral resolution of $1.55 \mu\text{m}$. These fabrication specifications in terms of phase bit-depth and lateral resolution are compatible with commercial two-photon polymerization-based 3D nanoprinters or lithography-based nanofabrication technologies. Especially wafer-scale high-throughput fabrication of such diffractive designs using high-purity fused silica (HPFS) that has an ultra-low loss with high thermal stability, as demonstrated in recent work⁴⁹, holds promise for mass-scale fabrication of cascable diffractive waveguides operating in the near-infrared spectrum^{42,50}.”*

Supplementary Fig. S6. Testing results of a single-mode diffractive waveguide operating at 1550 nm, designed with varying levels of phase quantization bit depths. (a) Input field, output field and diffractive layer phase profiles corresponding to different phase quantization bit depths. (b) Coupling and energy efficiency values of diffractive waveguides designed with different phase quantization bit depths.

6. The testing results of the bent diffractive waveguides at $K = 5$ are different for coupling and energy efficiency between Supplementary Figures S4 and S5. It would be necessary to clarify this.

-- We thank the referee for pointing out this discrepancy in the coupling efficiency and energy efficiency results for different configurations of bent diffractive waveguides. The difference was due to the different number (Q) of modes used, as defined in Equation 1 of the main text. This has been corrected in the revised SI file, i.e., $Q=6$ has been used for both **Supplementary Fig. S8** and **Supplementary Fig. S9**. The figures have been accordingly updated with the corrected results, and the discrepancy has been resolved.

Reviewer 2:

The authors use deep learning to develop diffractive patterns that can be used as an alternative to traditional refractive waveguides that rely on refractive index contrast. This framework is also capable of producing waveguides with specialized functionalities such as propagation rotation, mode splitting and filtering, and polarization maintenance. The authors compare their diffractive waveguide with a traditional square refractive waveguide showing low propagation loss and high coupling efficiency. Their work benefits from the lack of material dispersion in diffractive optics. In my judgement, this

demonstration and methodology for developing diffractive optics are significant additions to the literature and will have interest among a wide readership. Overall, the work provides extensive demonstrations of the benefits of the diffractive waveguide system and makes a compelling case for its use over conventional systems. Nonetheless, it is my assessment that the work would benefit from a few additional experiments and evidence as outlined below. Further, I believe the manuscript should also clarify a few points listed below to improvement the scientific quality the material.

-- We sincerely thank the referee for their valuable assessment and constructive feedback, which helped us further enhance the quality and clarity of our manuscript.

1. A key advantage of refractive processes over diffractive ones is higher efficiency. Can the authors provide a quantification of the energy loss from their diffractive system compared to the equivalent refractive dielectric square waveguide? While the amplitude and phase profiles present only small relative errors, energy loss is expected to be higher in the authors' diffractive system, so a quantification of that would be a valuable addition.

-- We thank the referee for raising this important point. In our study, we evaluated the performance of diffractive waveguides using both **coupling efficiency** and **energy efficiency**, as reported in **Fig. 2b**:

Figure 2. Comparative analysis for internal and external generalization ability of the diffractive waveguide. (a) Intensity and phase profiles of the input and output fields of the diffractive waveguide. Performance was evaluated using

two sets of spatial modes: internal generalization set $\{M_0, M_1, \dots, M_{19}\}$, which was used during the training stage, and external generalization set $\{M_{20}, M_{21}, \dots, M_{39}\}$, which was never used during the training. (b) Coupling efficiency and energy efficiency of the transmitted spatial modes $\{M_0, M_1, \dots, M_{39}\}$

For traditional dielectric waveguides, intrinsic attenuation losses can arise from, e.g., scattering and absorption. However, in our analysis, the dielectric waveguides used for comparison were assumed to have ideal refractive index profiles without any material defects and structural inhomogeneities. This assumption effectively eliminates various sources of losses for the standard dielectric waveguides considered in our comparison. Under these conditions, coupling and energy efficiencies are set to 100% for the traditional dielectric waveguides, serving as a reference baseline for evaluating the performances of our diffractive waveguide designs. These points have been incorporated into the **Results** section of our revised manuscript, as quoted below:

“... In our analysis, the dielectric waveguide used for comparison was assumed to have an ideal refractive index profile, without any material defects and structural inhomogeneities. This assumption effectively eliminates various sources of losses for the standard dielectric waveguides considered in our comparison. Under these conditions, coupling and energy efficiencies are set to 100% for the traditional dielectric waveguides, serving as a reference baseline for evaluating the performances of our diffractive waveguide designs.”

In Figure 1(e), the output fields from the square dielectric waveguide and the diffractive waveguide are also compared, with the relative errors displayed (at the bottom), which are negligible. The output fields of the traditional dielectric waveguide were calculated by full-vectorial finite-difference method, serving as the ground truth for evaluating the output fields of the diffractive waveguide.

Furthermore, to account for material absorption in experimental scenarios, we have introduced an additional analysis, as shown in the **newly added Supplementary Fig. S3**, where the training process incorporated material losses. The comparison between diffractive waveguides trained with and without material absorption reveals that coupling efficiency and energy efficiency can be improved significantly by taking into account material-induced losses in the training process.

Accordingly, a new figure, i.e., **Supplementary Fig. S3**, was added to the revised **Supplementary Information** and these new analyses were added to the **Results** section, quoted below:

*“... Additionally, to account for material absorption in practical implementations, we incorporated energy absorption into the training process, as shown in **Supplementary Fig.S3**. Compared to designs trained without considering material absorption, those optimized with absorption exhibited improved efficiency when tested with absorbing diffractive materials, demonstrating the benefit of accounting for such factors in the training loss function and optimization process.”*

Supplementary Fig. S3. Testing results of the diffractive waveguide models trained with and without incorporating material absorption into the training process. Both diffractive waveguide models were tested with an absorbing material; we assumed $\lambda = 0.75$ mm and the complex refractive index of the diffractive layers was assumed to be $1.72 + j0.03$, which corresponds to a commonly used 3D printing material (VeroBlack, Objet30 Pro, Stratasys). Lower loss materials can be used to further improve the energy efficiency of these diffractive waveguide designs

2. In Figure 2, the authors provide a visualization of the input and output fields generated from their model. There is a large drop in the coupling efficiency for modes M30-37. This drop casts doubt on the model's ability for external generalization as the decrease occurs primarily for the modes that were not used in the training set. The authors should explain this drop in coupling and provide evidence of whether this a physical result, or an artifact of their model's limitations.

-- We thank the referee for bringing up this concern about the generalization ability of the diffractive waveguide. As noted, the drop in coupling efficiency for modes $M_{30} \sim M_{37}$ in Fig. 2 corresponds to the modes that were **not included** in the training dataset. This behavior does not reflect a fundamental limitation of the model itself, but rather highlights the importance of matching the training mode set to the desired working conditions.

In waveguide-related applications, the working conditions, including the illumination wavelength, desired spatial modes and propagating distance, are typically determined a priori. Under such conditions, the diffractive waveguide can be specifically designed and optimized to handle the complete set of relevant and desired modes of interest. To further support this point, we have added **a new Supplementary Fig. S2** evaluating the effect of training set composition on model performance. In this new analysis, five diffractive waveguide models were trained with: I. 20 lower order modes ($M_0 \sim M_{19}$, same as Fig. 2); II. all 40 modes ($M_0 \sim M_{39}$); III. 20 higher order modes ($M_{20} \sim M_{39}$); IV. 10 a mixed set of lower order modes $M_0 \sim M_9$ and higher order modes $M_{30} \sim M_{39}$; V. an alternating pair based sampling: $\{M_0, M_1, M_4, M_5, M_8, M_9, \dots, M_{32}, M_{33}, M_{36}, M_{37}\}$. All models were tested with the full set of 40 spatial modes. **The second model, which was trained with the full set of modes, achieved the highest coupling efficiency values, while also retaining very high energy efficiency, as illustrated in Supplementary Fig.S2.**

These results confirm that the observed drop in coupling efficiency for $M_0 \sim M_{37}$ in Fig. 2 is not due to a limitation of the model architecture or the optimization process, but rather due to the absence of those modes in the training set, which can be significantly improved by including these modes in the training stage as illustrated in the new Supplementary Fig. S2.

These points and the new analyses have been added to the **Results** section, quoted below:

“... To further investigate the coupling efficiency drop observed for higher order modes, we analyzed the performance of diffractive waveguides trained on different spatial mode sets (see **Supplementary Fig.S2**). These results show that the coupling efficiency values are significantly improved when all the desired spatial modes of interest are known and accessible during training.”

Supplementary Fig. S2. Testing results of diffractive waveguide models trained with different sets of spatial modes. Five sets were used for training: I. 20 lower order modes ($M_0 \sim M_{19}$, same as in Fig. 2); II. all 40 modes ($M_0 \sim M_{39}$); III. 20 higher order modes ($M_{20} \sim M_{39}$); IV. a combination of 10 lower order modes $M_0 \sim M_9$ and 10 higher order modes $M_{30} \sim M_{39}$; V. two consecutive modes followed by skipping two, i.e., $\{M_0, M_1, M_4, M_5, M_8, M_9, \dots, M_{32}, M_{33}, M_{36}, M_{37}\}$. All the trained diffractive waveguide models were tested with the full set of 40 spatial modes

3. The authors claim that their system could be adapted for on-chip optical circuits. However, I am concerned about the feasibility of such applications given the massive reduction in feature size required for on-chip applications. Can the authors provide some estimated feature size required for an on-chip version of their design to support these statements?

-- We sincerely thank the referee for bringing up this important question. In **Supplementary Fig.S5 and S7**, we designed cascaded diffractive waveguides operating at 1550 nm and 532 nm, respectively. In those designs, the lateral feature size was set as $\sim \lambda/2$. To further support the practical viability of on-chip implementations, we conducted new simulations targeting the SMF-28 fiber mode profile at $\lambda = 1550 \text{ nm}$, using larger feature sizes and discretized phase modulation; SMF-28 is a single-mode optical fiber that is widely used in telecommunications. **As presented in the newly added Supplementary Fig. S6, >95% coupling efficiency and >95% energy efficiency can be achieved with 4-bit depth diffractive layers, with each layer having a lateral resolution of 1.55**

μm . These fabrication specifications in terms of phase bit-depth and lateral resolution are compatible with commercial two-photon polymerization-based 3D nanoprinters or lithography-based nanofabrication technologies. Especially wafer-scale high-throughput fabrication of such diffractive designs using high-purity fused silica (HPFS) that has an ultra-low loss with high thermal stability holds promise for mass-scale fabrication of cascaded diffractive waveguides operating in the near-infrared part of the spectrum.

These points, along with the new analysis reported in **Supplementary Fig. S6**, have been added to the **Results** section, quoted as:

*“... To explore the feasibility of implementations in the near-infrared part of the spectrum, we conducted simulations at 1550 nm wavelength for approximating the waveguiding behavior of SMF-28 fiber⁴⁸, which is a single-mode optical fiber that is widely used in telecommunications. As shown in **Supplementary Fig. S6**, >95% coupling efficiency and >95% energy efficiency can be achieved with 4-bit depth diffractive layers ($K = 2$), with each layer having a lateral resolution of 1.55 μm . These fabrication specifications in terms of phase bit-depth and lateral resolution are compatible with commercial two-photon polymerization-based 3D nanoprinters or lithography-based nanofabrication technologies. Especially wafer-scale high-throughput fabrication of such diffractive designs using high-purity fused silica (HPFS) that has an ultra-low loss with high thermal stability, as demonstrated in recent work⁴⁹, holds promise for mass-scale fabrication of cascaded diffractive waveguides operating in the near-infrared spectrum^{42,50}.”*

Supplementary Fig. S6. Testing results of a single-mode diffractive waveguide operating at 1550 nm, designed with varying levels of phase quantization bit depths. (a) Input field, output field and diffractive layer phase profiles

corresponding to different phase quantization bit depths. (b) Coupling and energy efficiency values of diffractive waveguides designed with different phase quantization bit depths

We have also added the following text related to this new **Supplementary Fig. S6**, quoted below: *“...the fabrication process of diffractive waveguides presents several challenges that must be addressed. One such challenge is the limited phase bit depth of the diffractive layers, often constrained by the resolution of the 3D fabrication technique employed. Additionally, issues such as physical misalignments can degrade performance. To mitigate these challenges, we can incorporate these limitations of the fabrication and alignment methods directly into the training process of the diffractive waveguide design. Specifically, the limited phase bit depth can be added as a design constraint during the training process such that the phase bit depth necessary for the effective operation of the diffractive waveguide can be reduced to e.g., 2-4 bits^{36,39-41}, as illustrated in **Supplementary Fig. S6**, which not only makes the fabrication process less demanding but also aligns well with the current manufacturing capabilities^{41,62}. Furthermore, a “vaccination” strategy can be adopted to enhance the resilience of diffractive waveguides to unknown and random misalignments; this can be achieved by randomly shifting the diffractive layers of the waveguide design around their ideal positions during the training^{36,37,39,40,44,54}. Such a vaccination strategy would effectively reduce the independent degrees of freedom within the diffractive waveguide design, which can be mitigated using wider or deeper diffractive unit architectures.”*

4. The demonstrating of spatial mode filtering is quite convincing and impressive; however, in my opinion, it would benefit from benchmarking to conventional spatial mode filters in terms of rejection ratios and coupling efficiencies.

-- We thank the reviewer for the positive and encouraging feedback, as well as the thoughtful suggestions. To address these points, we have added and revised the following text and paragraphs, quoted below:

“...Compared to conventional spatial mode filtering systems, such as asymmetric Y-junctions⁵⁹, multi-mode interference (MMI) couplers⁵⁸, and long-period gratings⁵⁹, our diffractive platform offers a number of distinct advantages. These conventional devices exhibit excellent performance in terms of coupling efficiency and mode rejection. However, extending their operation across a broad spectral band typically requires careful dispersion engineering, precise refractive index contrast, and complex fabrication processes. In contrast, our diffractive waveguide designs can be extended to operate simultaneously at multiple wavelengths and polarization/mode combinations, performing a different desired function at each wavelength channel and polarization state or mode. For example, such a capability can be achieved by incorporating each wavelength channel with its associated desired task into the loss function during the deep learning-based training process⁶². Notably, the adaptation to multiple wavelength channels does not require material dispersion engineering. Instead, these layers can be constructed from low-loss materials, regardless of the dispersion properties of the diffractive materials⁶², simplifying the design process and eliminating the dependency on the dispersion properties of the materials used. Therefore, the entire process of modal dispersion engineering within a diffractive waveguide can be handled by deep learning-based loss function optimization, as opposed to material engineering.”

*“...To further demonstrate the unique capabilities of these mode filtering diffractive waveguide designs, a multi-wavelength mode filter was developed, as shown in **Fig. 6a**. This diffractive*

waveguide was optimized to selectively transmit different sets of spatial modes at certain illumination wavelengths. Specifically, the spatial modes in $S_{\lambda_1} = \{M_0, M_{11}\}$, $S_{\lambda_2} = \{M_6 \sim M_{11}\}$ and $S_{\lambda_3} = \{S_{16} \sim S_{19}\}$ can only transmit at the corresponding illumination wavelengths $\lambda_1 = 0.7 \text{ mm}$, $\lambda_2 = 0.75 \text{ mm}$ and $\lambda_3 = 0.8 \text{ mm}$, respectively. All other modes were filtered out through this multi-wavelength mode filtering diffractive waveguide. Following a similar training strategy used for its monochromatic counterpart, the multi-wavelength design was effectively optimized to select distinct sets of modes at different illumination wavelengths. **Figure 6b** illustrates the input and output patterns across different wavelengths. The resulting coupling efficiency and the energy efficiency values, shown in **Fig. 6c**, further highlight the success of the multi-wavelength filter function, where the desired, transmitted modes maintain high quality, and the rejected modes exhibit low coupling and energy efficiencies within the targeted wavelengths. By adjusting the loss function during the training phase, diffractive waveguides with tailored functionalities for more advanced mode filtering across specific mode orders and wavelengths can be achieved.”

“...In a more general scenario, spatial modes with multiple wavelengths within a single input aperture can also be demultiplexed into different channels. To demonstrate this capability, we report a multi-wavelength mode splitting diffractive waveguide which can split the input aperture into 2×2 outputs based on the order of the spatial modes and illumination wavelengths. As illustrated in **Fig. 8a**, the input modes in two different illumination wavelengths entering the input aperture were separated into four output channels based on their wavelength and spatial mode order: at illumination wavelength $\lambda_1 = 0.7 \text{ mm}$, modes in $S_{CH1} = \{M_0 \sim M_{11}\}$ and $S_{CH2} = \{M_{12} \sim M_{19}\}$ were directed into Channel 1 and Channel 2, respectively; at illumination wavelength $\lambda_2 = 0.8 \text{ mm}$, modes in $S_{CH3} = \{M_0 \sim M_7\}$ and $S_{CH4} = \{M_8 \sim M_{19}\}$ were directed into Channel 3 and Channel 4, respectively. The trained model was blindly tested with different orders of optical modes at these two illumination wavelengths individually, shown in **Fig. 8b**. At λ_1 illumination, the multi-wavelength mode splitting diffractive waveguide successfully guided all the modes in S_{CH1} into Channel 1 and the other modes in S_{CH2} into Channel 2 while exhibiting no apparent crosstalk with Channels 3 and 4. Similarly, Channels 3 and 4 were able to receive their desired set of modes at illumination wavelength λ_2 with no evident leakage into Channels 1 and 2. Furthermore, we quantified the coupling efficiency and the energy efficiency of these four output channels for all spatial modes of interest in **Fig. 8c**, showing a good mode splitting performance and suppression of unwanted modes in each output channel.”

“...In addition to our monochromatic experimental results, we also designed a multi-wavelength mode filtering diffractive waveguide to pass different sets of modes at different illumination wavelengths, as depicted in **Fig. 14a**. We followed the same process mentioned in the monochrome design while involving two illumination wavelengths, $\lambda_1 = 0.7 \text{ mm}$ and $\lambda_2 = 0.8 \text{ mm}$. Once the model converged, the diffractive layers were 3D printed and aligned, as shown in **Fig. 14b**. We numerically calculated the coupling efficiency and the energy efficiency at the output plane for spatial modes at different illumination wavelengths, as shown in **Fig. 14c**. This multi-wavelength mode filtering diffractive waveguide design successfully passed $\{M_2, M_3\}$ at the illumination wavelength $\lambda_1 = 0.7 \text{ mm}$ and passed $\{M_{12}, M_{13}\}$ at the illumination wavelength $\lambda_2 = 0.8 \text{ mm}$, while rejecting all other modes at both wavelengths. The fabricated multi-wavelength mode filtering diffractive waveguide was tested experimentally using four different phase-only apertures. These 3D-printed phase apertures, as illustrated in the first and second columns of

Fig. 14d, were used to generate specific structured optical fields at different illumination wavelengths, shown in the third and fourth columns of **Fig. 14d**, which contained different ratios of optical modes. The first phase aperture could excite the corresponding target modes for both illumination wavelengths, the second and third apertures could individually excite the target modes at different wavelengths, while the optical field produced by the fourth phase aperture did not include any target modes. The outputs of this 3D fabricated multi-wavelength mode filtering diffractive waveguide were numerically simulated and experimentally measured using the same setup. Cross correlation operation was used to evaluate the similarity between the experimental results and the simulated/numerical results (see the Methods section), which showed a good agreement between the two, experimentally demonstrating the effective mode filtering functionality of the 3D printed multi-wavelength mode filtering waveguide.”

5. My main concern with this work is its novelty compared to the extensive literature on diffractive optics and meta lenses. Many of the functionalities demonstrates here mode splitting, and spatial mode filtering have been demonstrated in prior works (Khonina, Svetlana N., et al. "Advancements and applications of diffractive optical elements in contemporary optics: A comprehensive overview." *Advanced Materials Technologies*(2024): 2401028.).

Thus, the main novelty seems to be the deep learning algorithm developed by the authors and the modularity of their system. My suggestion is that the authors clarify the novelty and uniqueness of their work by comparing it to similar prior works. Any experiments demonstrating this novelty or an improvement in optical parameters would bolster the case for diffractive waveguides.

Following the above recommended changes, I judge that the manuscript will be ready for publication.

-- We thank the reviewer for raising these important points regarding existing literature on diffractive optics and meta lenses. To address these points, we have added and revised the following text and paragraphs, quoted below:

“...In contrast to earlier diffractive optics and metasurface-based studies that generally focus on isolated or single-function devices, our work presents a unified, diffractive waveguide platform that supports a broad range of integrated functionalities. Notably, the presented diffractive waveguide platform enables spatial and spectral mode filtering and mode splitting designs, as well as mode-specific polarization maintenance, offering a level of control that is challenging to achieve with conventional designs. Another important distinction between our approach and prior work lies in the cascadability^{55–57} of diffractive designs at high numerical apertures, e.g., NA=1. Most metasurface architectures have approximation errors in their forward models at such a high numerical aperture, which can introduce cascaded errors that will build up as the length of the diffractive waveguide system increases. As a result, they are typically restricted to implementing a single function or an isolated task without a deeper cascaded architecture. In contrast, our diffractive waveguide platform operates by processing all the propagating modes within free-space, i.e., with an NA of 1 in air, enabling modular cascading of transmissive diffractive surfaces, each contributing to a unified, end-to-end functionality. This task-specific cascadability, supported by our deep learning-based optimization and design pipeline, allows us to realize a wide range of spatial, spectral, and polarization-specific operations within a single integrated framework—highlighting another key aspect of our work’s novelty and utility beyond existing diffractive and metasurface architectures.

“...our diffractive waveguide designs can be extended to operate simultaneously at multiple wavelengths and polarization/mode combinations, performing a different desired function at each wavelength channel and polarization state or mode. For example, such a capability can be achieved by incorporating each wavelength channel with its associated desired task into the loss function during the deep learning-based training process⁶². Notably, the adaptation to multiple wavelength channels does not require material dispersion engineering. Instead, these layers can be constructed from low-loss materials, regardless of the dispersion properties of the diffractive materials⁶², simplifying the design process and eliminating the dependency on the dispersion properties of the materials used. Therefore, the entire process of modal dispersion engineering within a diffractive waveguide can be handled by deep learning-based loss function optimization, as opposed to material engineering.”

The deep learning-based task-specific optimization algorithm used in our design framework is a critical enabler of the proposed diffractive waveguide platform, allowing us to jointly optimize multiple objectives, including spectral selectivity, spatial mode control, and polarization behavior. By incorporating multiple tasks into the loss function during training, our approach enables the design of devices that can simultaneously operate at multiple wavelengths, with each wavelength channel performing a distinct, application-specific function—a level of functional control that is not easily achievable using conventional waveguide or diffractive design methods. These points are also addressed in the current version of the manuscript, as quoted below:

“...our diffractive waveguides use cascaded phase modulation through spatially optimized diffractive layers. This simplifies the design of task-specific waveguides, where the training loss function of a cascadable diffractive waveguide unit can be optimized for different goals covering various spectral, spatial and polarization features of interest, without the need for material dispersion engineering.”

*“...Unlike conventional dielectric material-based solid-core waveguides, the diffractive waveguides presented here are based on cascadable discrete diffractive layers that periodically modulate the phase structure of light without the need for any material dispersion engineering. **These cascaded structured diffractive layers surrounded by air (or specific types of gas or liquid, if desired) give us new degrees of freedom to create and assemble task-specific waveguide topologies composed of repeating dielectric surfaces designed by deep learning. With minimal adjustments made according to the desired task, these diffractive waveguides can be optimized, offering a standardized approach to task-specific waveguide design. Additionally, these waveguides can be scaled to operate across different parts of the electromagnetic spectrum without the need to redesign their structures or material engineering, by simply scaling the diffractive feature size proportional to the wavelength of interest, forming a versatile waveguide platform, enhancing the flexibility of conventional designs⁴². Finally, diffractive waveguides can also control and multiplex the polarization states of guided modes by the integration of a 2D polarizer array within the diffractive unit design^{38,45}. These diffractive waveguides will inspire further research and development, proving beneficial for a variety of applications in e.g., telecommunications, imaging, sensing and spectroscopy.”***